# Impacts of fertilization on grassland productivity and water quality across the European Alps under current and warming climate: insights from a mechanistic model

Martina Botter[1], Matthias Zeeman[2], Paolo Burlando[1], Simone Fatichi[3]

[1]Institute of Environmental Engineering, ETH Zurich, Zurich, Switzerland

[2] Karlsruhe Institute of Technology, Institute of Meteorology and Climate Research, Atmospheric Environmental Research, Garmisch-Partenkirchen, Germany

[3] Department of Civil and Environmental Engineering, National University of Singapore, Singapore

*Correspondence to*: Martina Botter (botter@ifu.baug.ethz.ch)

**Abstract.** Alpine grasslands sustain local economy providing fodder for livestock. Intensive fertilization is common to enhance their yields, thus creating negative externalities on water quality that are difficult to evaluate without reliable estimates of nutrient fluxes. We apply a mechanistic ecosystem model, seamlessly integrating land-surface energy balance, soil hydrology, vegetation dynamics, and soil biogeochemistry aiming at assessing the grassland response to fertilization. We simulate the major water, carbon, nutrient, and energy fluxes of nine grassland plots across the broad European Alpine region. We provide an interdisciplinary model evaluation confirming its performance against observed variables from different datasets. Subsequently, we apply the model to test the influence of fertilization practices on grassland yields and nitrate ($NO_3^-$) losses through leaching under both current and modified climate scenarios.

Despite the generally low $NO_3^-$ concentration in groundwater recharge, the variability across sites is remarkable, mostly, but not exclusively, dictated by elevation. In high-Alpine sites short growing seasons lead to less efficient nitrogen (N) uptake for biomass production. This combined with lower evapotranspiration rates results in higher amounts of drainage and $NO_3^-$ leaching to groundwater. Scenarios with increased temperature lead to a longer growing season characterized by higher biomass production and, consequently, to a reduction of water leakage and N leaching. While the inter-site variability is maintained, climate change impacts are stronger on sites at higher elevations.

The local soil hydrology has a crucial role in driving the $NO_3^-$ use efficiency. The commonly applied fixed threshold limit on fertilizer N input is suboptimal. We suggest that major hydrological and soil property differences across sites should be considered in the delineation of best practices or regulations for management. Using distributed maps informed with key soil and climatic attributes or systematically implementing integrated ecosystem models as shown here can contribute to achieving more sustainable practices.

## 1 Introduction

Alpine grasslands are a vital resource for the European economy. They provide a variety of ecosystem services such as maintaining biodiversity, protecting soil, and offering recreational functions (Sala and Paruelo, 1997; Lamarque et al., 2011; Schirpke et al., 2017). Above all, they sustain economy providing fodder for livestock. To this purpose high yields are desired, as they correspond to increasing income for farmers, hence large amounts of fertilizers are applied every year. One of the most important components of fertilizers is nitrogen. On the one hand nitrogen plays a crucial role as a nutrient contributing to the

grasslands' growth. On the other hand, it is one of the main responsible component for environmental pollution from agriculture. Nitrogen is lost into the environment either through gaseous emissions of $N_2O$, $N_2$ and $NH_3$ (Amon et al., 2006; Ibraim et al., 2019; Schlingmann et al., 2020), or as $NO_3^-$, either through surface runoff or as leaching to groundwater. Well-known water quality issues, such as eutrophication, result directly from intensive management of grasslands (e.g., Heathwaite, 1995; Peukert et al., 2014). In order to tackle the problem of $NO_3^-$ losses to the environment, the European Union (EU)

approved the Nitrate Directive 91/676/EEC (EEC, 1991), which imposes limits on the yearly load of fertilizers and restricts the fertilization season. In this frame, the EU countries should pursue grassland management strategies designed with the intention of maximizing the grassland yields without damaging the environment. The task is particularly daunting given the spatial heterogeneity across the wide Alpine grassland areas and given the threat of climate change, which is likely to remarkably impact the nitrogen cycle (Wang et al., 2016). Elevation is often a major constrain for the management in Alpine

environment, being low-elevation sites generally more prone than high-elevation sites to intensive management, for both climatological and pragmatic reasons. Since the EU Nitrate Directive has been in effect, many studies attempted to quantify the $NO_3^-$ fluxes with various methods (e. g., Kronvang et al., 2009; Groenendijk et al., 2014). However, according to a questionnaire submitted to grassland experts across EU countries, Norway and Switzerland (Velthof et al., 2014), "non-academic" quantification of $NO_3^-$ fluxes is still mainly based on expert estimates or on prescribed values. Rarely models are

applied. When they are, the most common approach is the simple empirical feed balance method, where grassland yields are estimated using statistical data on feed availability for ruminants and their feed requirements. The nutrients balance is consequently estimated applying values derived from literature or direct measurements of nitrogen content in sampled grass. More complex modelling approaches for the quantification of nitrate losses from agricultural activity are emerging. Some European Projects have been promoted with the goal of assessing the state-of-the-art modelling tools in view of a harmonized

assessment procedure across Europe. For example, the EUROHARP Project (Kronvang et al., 2009) compares available distributed models applied in different European countries. The analyzed models span from conceptual models such as NLES_CAT (Simmelsgaard and Djurhuus, 1998) and MONERIS (Behrendt et al., 2003) to more complex and process-oriented

models like SWAT and NL-CAT, born from the integration of ANIMO (Groenendijk et al., 2005), SWAP (Kroes and Dam, 2003), SWQN (Smit et al., 2009) and SWQL (Siderius et al., 2009). Another example is the GENESIS Project (Groenendijk et al., 2014), which compares 1-D mechanistic models integrating hydrology, vegetation dynamics, and soil biogeochemistry. The performance of some models such as ARMOSA (Perego et al., 2013), CoupModel (Jansson, 2012) and EPIC (Williams et al., 1984; Sohier et al., 2009) are compared to simulated nitrate leaching rates from lysimeters. What emerges from these studies is that mechanistic models are currently not used for the purpose of supporting management because they are thought to be difficult to operate, require a large amount of input data, and require complex parameterization (see also discussions in Fatichi et al., 2019). However, when the above limitations are overcome, mechanistic models could provide insights on biomass and nutrients budget quantification that would be too difficult to obtain otherwise. They can constrain the description of soil biogeochemical processes by means of mass and stoichiometric constraints better than empirical approaches providing a larger predictive power ( Moorhead et al., 1996; Wieder et al., 2013; Manzoni et al., 2016).

The $NO_3^-$ leaching depends on the overall N cycle, which results from the interplay of various processes. Hence, mechanistic models reproducing the whole N cycle require a tight integration with models or modules dealing with land-surface exchanges of energy and water, vegetation dynamics, soil hydrological and biogeochemical processes and hydrological transport, in other words what are called ecosystem or terrestrial biosphere models (Fatichi et al., 2016).

Different ecosystem models have been developed, but they rarely integrate all the above mentioned modules, or treat those with a similar level of complexity. For example, models that were born as tools for the estimation of the greenhouse gases emissions tend to simplify the soil hydrology using bucket-type approaches. Some examples of this kind of models are CERES-EGC (Gabrielle et al., 1995; Gabrielle and Kengni, 1996; Hénault et al., 2005), DAYCENT ( Parton et al., 1998; 2015; Del Grosso et al., 2000; 2002), STICS (Brisson et al., 1998; 2002; 2003) and DNDC (Li et al., 2000; 2012), although efforts have recently been made to enhance the hydrological module of the latter (Smith et al., 2020). In other models, the soil hydrology module is represented with much higher detail than the soil biogeochemistry module, as is the case for HYDRUS-1D/2D/3D ( Tafteh and Sepaskhah, 2012; Phogat et al., 2013) or DAISY (Hansen, 1990; 2002). More recently, ecosystem models were built combining different models, each highly specialized in a specific compartment of the ecosystem. The result leads to fully-integrated models such as ARMOSA (Perego et al., 2013), which is the combination of the hydrological model SWAP (Van Dam, 2000), STAMINA (Ferrara et al., 2010; Richter et al., 2010) for simulating the crops dynamics, and SOILN (Bergström et al., 1991) to represent the soil carbon and nitrogen cycles. The model SIM-STO (Pütz et al., 2018) was also created from the merging of SIMWASER (Stenitzer, 1988) and STOTRASIM (Feichtinger, 1998), respectively focused on water fluxes and nitrogen dynamics. RT-Flux-PIHM (Bao et al., 2017) combines a reactive transport (RT) model, a land-surface model (Noah LSM, Chen et al., 2001; Shi et al., 2013) and the hydrological model PIHM (Shi et al., 2013). These models are usually performing well when applied with targeted applications. In most of the cases they are validated only against the data concerning one specific module that is the most important to solve the problem at hands. Hardly ever the models are concurrently evaluated against vegetation and soil water dynamics, energy and biogeochemical fluxes. Here, we push the

envelope of integrated models by applying the mechanistic ecosystem model T&C-BG, which simulates the interactions among vegetation dynamics, soil biogeochemistry, and hydrological fluxes. T&C-BG was recently developed (Fatichi et al., 2019) as a single fully integrated model and therefore does not require many of the pragmatic simplifications necessary to externally couple different models. We evaluate the model capabilities by applying a "common" (non-site specific) parametrization to nine managed grassland sites across the broad alpine region, derived from previous plot-scale studies (Fatichi et al., 2014; Fatichi and Pappas, 2017). First, we evaluate the model against hydrological, energy, vegetation, and soil biogeochemical observations according to the available data. Then, we run numerical experiments to estimate the grass yields and the losses of $NO_3^-$ under different fertilization regimes and compare results across the sites. Ultimately, we simulate a climate change sensitivity test for each management scenario modifying atmospheric $CO_2$ concentrations of +250 ppm and air temperature of +3 °C, to compare results with outcomes from the simulations under present climate. Specifically, we investigate and answer the following questions:

(1) Is a mechanistic ecosystem model provided with a "common" parameterization for Alpine managed grasslands able to reliably represent the ecosystem dynamics across 9 sites?

(2) How do grass productivity and leaching of $NO_3^-$ change in response to different fertilization scenarios across the broad Alpine region?

(3) Can mechanistic models provide insights for legislators setting guidelines for management also in view of changing climatic conditions?

Answering these questions, beyond allowing the assessment of a state-of-the-art ecosystem model in reproducing multiple aspects of ecosystem functioning, also provides quantitative information that can be used to delineate best grassland management practices.

## 2 Study sites and methods

### 2.1 Study sites and instrumentation

The study sites are located in European Alps and cover an altitude gradient ranging between 393 m a.s.l and 2160 m a.s.l. We used grassland sites in the Alpine region for which at least eddy covariance flux tower observations were available. For a subset of sites, additional observations about leaf area index, soil moisture, grass productivity and water and nitrogen leaching were also available. Specifically, we used the sites of Torgnon (IT-Tor) and Monte Bondone (IT-MBo) in Italy, Oensingen (CH-Oe1), Chamau (CH-Cha) and Früebüel (CH-Fru) in Switzerland, Neustift (AT-Neu) in Austria and Fendt (DE-Fen), Rottenbuch (DE-RbW) and Graswang (DE-Gwg) in Germany. The mean annual precipitation (MAP) across sites ranges between 850 mm and 1627 mm while mean annual temperature varies between 2.9 and 9.5 °C following the elevation gradient (Table 1). All these sites are permanent grasslands ecosystems, mainly used for fodder production (Ammann et al., 2007;

Hammerle et al., 2008; Vescovo and Gianelle, 2008; Kiese et al., 2018). Apart from the site IT-Tor, which is unmanaged, the other grassland sites are managed with regular fertilizer applications and harvested with grass cuts at standard height (0.07 m). Each site is equipped with a flux tower providing time series of micrometeorological variables, as well as energy (e.g., net radiation, latent heat, sensible heat) and carbon dioxide fluxes by means of the Eddy Covariance (EC) method (Aubinet et al., 2012). The flux tower data for all the sites but the German ones are retrieved from the FLUXNET-2015 database (Pastorello et al., 2020). German sites (DE-Fen, DE-RbW, DE-Gwg) belong to the Pre-Alpine Observatory of the TERENO network (Zacharias et al., 2011; Pütz et al., 2016; Kiese et al., 2018) where lysimeters are also installed in proximity of the flux-tower site (Fu et al., 2017, 2019; Zeeman et al., 2017; 2019).

For the stations DE-Fen, DE-RbW and DE-Gwg annual totals of lysimeter evapotranspiration (ET) and water leakage are retrieved for the years 2012-2014 from Fu et al., (2017, 2019). The same study provides annual DOC-C and nitrate-N (N-$NO_3^-$) concentrations in the groundwater recharge. The ScaleX campaign of 2015 (Wolf et al., 2017; Zeeman et al., 2019) provided additional data concerning harvested biomass and its carbon and N content.

## 2.2 T&C-BG model

We simulate the grassland ecosystem dynamics in each site with the fully-coupled terrestrial biosphere/ecosystem model T&C-BG, introduced by Fatichi et al. (2019), who provided a complete model description. T&C-BG combines a well-tested ecohydrological model (T&C, e.g., Fatichi et al., 2012a, 2012b; Fatichi et al., 2014, 2015; Manoli et al., 2018; Mastrotheodoros et al., 2020) with a soil biogeochemistry module, which represents soil carbon and nutrient dynamics as well as plant mineral nutrition and bilateral feedbacks between nutrient availability, plant growth and soil mineralization processes. The T&C-BG model inputs include hourly flux-tower observations of microclimate (i.e., precipitation, air temperature, wind speed, relative humidity, incoming shortwave and longwave radiation, atmospheric pressure) and resolves the principal land-surface energy exchanges (i.e., net radiation, sensible and latent heat) which are interconnected with hydrological processes, such as evaporation and transpiration, infiltration, runoff, saturated and unsaturated zone water dynamics, groundwater recharge, as well as snow and ice hydrology dynamics. The soil hydrology module solves the 1-D Richard's equation in the vertical direction and uses a heat diffusion solution to compute the soil temperature profile. A soil-freezing module has been also recently introduced (Yu et al., 2020). The vegetation module computes photosynthesis, respiration, vegetation phenology, carbon and nutrient budget including allocation to different plant compartments and tissue turnover.

The management of vegetation such as grass cuts and fertilization can be prescribed as part of model inputs. The soil biogeochemistry module accounts for the carbon and nutrient budgets of litter and soil organic matter. Mineral nitrogen (N), phosphorous (P), and potassium (K) budgets and the nutrient leaching are also simulated. This study is particularly focused on $NO_3^-$. Nitrate pool in the soil depends on the net immobilization/mineralization fluxes, nitrogen uptake, nitrogen leaching, and nitrification/denitrification fluxes, with the latter that are simulated with empirical functions of the amount of $NO_3^-$ and

environmental conditions (detailed description in Fatichi et al., 2019). $NO_3^-$ transport process is not solved in the soil column, but leaching is assumed to occur at the bottom of the soil column and it is proportional to the water leakage and bulk $NO_3^-$ concentration in soil water in the soil biogeochemical active zone (Fatichi et al., 2019).

The model can be run in a distributed topographically complex domain (e.g., Mastrotheodoros et al., 2019, 2020) but here it is employed in its plot-scale version, which simply solves 1-D vertical exchanges. In summary, based solely on meteorological inputs, soil and vegetation parameters, T&C-BG simulate prognostically all the other variables, from energy, carbon, and water fluxes, to vegetation biomass, up to soil nutrient mineralization and leaching. All these processes interact with each other.

## 2.3 Model parametrization

The model allows the simulation of different vegetation types, but in this case only grass is included. In order to keep results general enough and maximize future model transferability across grassland sites, we adopt a common parametrization for all of the nine sites with few exceptions justified by elevation dependent parameters such as threshold temperature for leaf onset. Vegetation parameter selection was based on previous experience with European grasslands (Fatichi et al., 2014; Fatichi and Pappas, 2017), soil biogeochemistry parameters are currently fixed for all sites in absence of more specific information (see discussion in Fatichi et al. 2019). The site-specific soil content of clay, sand, and organic matter in each site is provided as input parameter to the model, which internally computes the hydraulic soil parameters by means of pedo-transfer functions (Saxton and Rawls 2006). To make results comparable, across all the sites we assume a soil depth equal to 1.4 m, corresponding to the depth of lysimeters, which is discretized into 16 soil layers of increasing depth from the surface to the bedrock. The biogeochemistry active zone is fixed at 25 cm depth and the grass roots are assumed to be exponentially distributed with a maximum rooting depth of also 25 cm. We do not simulate the groundwater dynamics, but we compute the groundwater recharge as the water leakage at 1.40 m depth. The parameters selected for all simulations are detailed in Table S1 where the site-specific parameters are highlighted in bold.

In each site we force the model with the local meteorological conditions, while atmospheric $CO_2$ concentrations are assumed to follow the observed historical global trend (Keeling et al., 2009). Nutrients depositions in absence of local specific information for each site are set using global maps of recently observed values for nitrogen and phosphorus (Galloway et al., 2004; Mahowald et al., 2008; Vet et al., 2014).

Simulations corresponding to the described setup are used for evaluating the model performance before running the scenario analysis and they represent the reference simulations. We evaluate the model performance against an ensemble of diverse variables. First, we compare simulated and observed net radiation, latent heat, sensible heat, Gross Primary Production (GPP) and Net Ecosystem Exchange (NEE) using flux towers data. Second, we test the hydrological module comparing the effective soil saturation (e.g., normalized soil moisture) from the model and from local measurements. Third, we evaluate the vegetation module comparing the simulated biomass and/or Leaf Area Index (LAI) dynamics with data retrieved from literature wherever available (Ammann et al., 2007; 2009; Hammerle et al., 2008; Gilgen and Buchmann, 2009; Zeeman et al., 2010; Chang et al.,

2013; Finger et al., 2013; Filippa et al., 2015; Prechsl et al., 2015). Finally, we compare biogeochemical fluxes in terms of harvested nitrogen and carbon, and leaching of $NO_3^-$ and DOC. The evaluation of the soil biogeochemistry module is possible only for the sites DE-Fen, DE-RbW and DE-Gwg, equipped with lysimeters, where, however, differences in scale, soil properties, and timing of management between flux-tower footprint and lysimeters (Oberholzer et al., 2017; Mauder et al., 2018) exist.

Detailed manure input data for all the case studies were not available. To bypass this problem, whenever we did not have information about fertilization (i.e., in all sites except in the German ones), only for the reference simulations we assume that the cut grass is left on the ground. Such hypothesis is clearly unrealistic because it opposes the purpose of the grassland management (i.e., producing yields), but it guarantees a nearly closed nutrient cycle, thus performing the same function of fertilizers, but providing the most targeted fertilization possible. While aiming at testing the model performance on the baseline

scenario, such a hypothesis is preferred to assuming a fertilization rate for each site to avoid excessive or insufficient nutrient addition.

In DE-Fen, DE-RbW and DE-Gwg, the sites where both the flux tower and the lysimeters are co-located, grassland is managed slightly differently above the lysimeters and in the flux tower footprint. To validate model results we run multiple simulations, each fed with the corresponding management either of flux-tower or lysimeter plots.

Since the current initial conditions of the carbon and nutrients pools in the soil are unknown, as common in modeling studies, we spin-up carbon and nutrient pools running only the soil-biogeochemistry module for 1000 years using average climatic conditions with prescribed litter inputs taken from preliminary simulations with the soil-biogeochemistry module inactive. Then we used the spun-up carbon and nutrient pools as initial conditions for the hourly-scale fully coupled simulation over the period for which hourly observations are available. This last operation is repeated two times which allows reaching a dynamic

equilibrium. We compare available observation of soil C:N values retrieved from literature with the values resulting from such a spin-up process.

### 2.4 Numerical fertilization experiments and climate change scenario

We set up numerical experiments to test the response of the study sites to different manure application regimes under historical climate and with modified temperature and atmospheric $CO_2$ conditions potentially representative of the future. First, we

classify the sites based on elevation and managemen. We named pre-Alpine/Intensive sites located at elevations lower than 800 m a.s.l. and intensively managed (CH-Cha, CH-Oe1, DE-Fen, DE-RbW), Alpine/Extensive for sites with an altitude between 800 and 1000 m a.s.l. (DE-Gwg, AT-Neu, CH-Fru) and high-Alpine/Extensive sites above 1000 m a.s.l. extensively managed, which include the two grasslands of IT-MBo and IT-Tor. It is common practice among farmers to fertilize at the beginning of the growing season and after each cut. The number of cuts and manure applications along the year decreases with

elevation. Although the time intervals between cuts can be similar, the length of the winter dormant period ultimately determines growing season length and, in turn, the number of cuts, the volume of locally needed organic fertilizer and the

opportunity to bring that out into the field. We aligned the management strategy of our numerical experiments to these notions and used a classification of the sites into three groups. In our simulations, manure is applied 6, 4, and 2 times and the grass is cut 5, 3 and 1 times, respectively, in pre-Alpine, Alpine and high-Alpine sites. Grass is cut at a height of 0.07 m, following

common practice. Referring to literature (Ammann et al., 2007, 2012; Merbold et al., 2014; Fu et al., 2017) we identified a broad range of possible N yearly loads applied in grasslands. For pre-Alpine sites the simulated range spans between 50 kg ha[-1] yr[-1] (i.e., extensive management) and 500 kg ha[-1] yr[-1] (highly intensive management). This upper limit intentionally exceeds the actual management practices and allows us to analyze what can happen if fertilization loads are increased. We computed the corresponding single application amount of manure based on C:N of manure ratio equal to 8.9, which is reported for the

sites of DE-Fen, DE-RbW and DE-Gwg (Fu et al., 2017), and is also similar to values suggested in literature (Sommerfeldt et al., 1988; Nyamangara et al., 1999) or slightly lower (Zhu, 2007; Kumar et al., 2010). The input of P and K is assumed to be proportional to the N input based on assigned N:P and N:K values guaranteeing non P- or K-limited conditions to the system. The specific manure amount computed for pre-Alpine sites is applied for Alpine and high-Alpine sites at the corresponding lower frequencies. The resulting annual loads in all the scenarios are reported in Table 2. For each fertilization scenario, in

each site, we spin-up the system running the soil-biogeochemistry module for 1000 years under average climatic conditions and the specific management scenario. Thus, the initial conditions of each simulation correspond to the final state of the spin-up simulation run for each specific site and management scenario. Finally, we run each management scenario in each site under both historical and modified climate scenario, representing the latter with an increase of atmospheric $CO_2$ concentration of 250 ppm and an increase of air temperature of +3°.


In the result analysis, we focus our attention on the resulting N contribution to grass productivity and in the N lost as leaching at the bottom of the soil profile. The former represents one of the positive gain from agriculture as economic activity and the latter a negative externality into the environment, as most $NO_3^-$-leaching will ultimately reach the groundwater storage and the rivers. We evaluate the different management strategies bringing together the two indicators in an index computed as the ratio

of harvested-N to N- $NO_3^-$ leachate. The higher the value of the index the most favorable is the management strategy for both farmers and environment. Thus, we interpret this index as a proxy of the efficiency of the grassland in profiting from the N added in fertilization. We also analyze potential relations between such efficiency index and site-specific characteristics, such as the elevation and the percentage of precipitation that becomes groundwater recharge. We further assess the impact of increased atmospheric $CO_2$ concentration and  temperature on the short term by comparing the results of the historic and

modified climate scenarios. We compare variables such as water leakage, harvested N, N leaching and we analyses their implications for the efficiency index.

# 3 Results

## 3.1 Model evaluation

The performance of the model in representing the energy and carbon fluxes measured from the flux towers is good across all the sites. The $R^2$ values of the model/observations comparison are reported in Table 3 and further goodness of fit metrics are reported in Table S2. The seasonality of the energy and carbon fluxes is well represented across all the study sites, as illustrated by the pattern of latent heat shown in Figure 2 and the patterns of H, GPP, and NEE reported in Figure S1 and Figure S2 and Figure S3 respectively. One exception is the NEE seasonality in sites where top soil freezing is simulated as IT-MBo, with NEE peaking later in the model.

We show the comparison between the observed and the simulated effective saturation of the soil (Figure 3) to evaluate the hydrological dynamics. The intra-annual pattern is generally well captured by the model. The average coefficient of determination $R^2$ equals 0.49 and the average RMSE is 0.10. Results highlight the pedo-climatic differences across the Alpine region. Some sites tend to reach field capacity quite easily (e.g., DE-Fen, DE-RbW and DE-Gwg), in other sites effective saturation is often below 50% (CH-Cha, CH-Oe1, AT-Neu and CH-Fru). The high-Alpine sites IT-Tor and IT-MBo fall in this second case, but with pronounced peaks of saturation when snow melt occurs.

A summary of the simulated site-specific variables in terms of water, energy and carbon fluxes is shown in Table 4. The total net radiation does not vary consistently with elevation. As a result of temperature constraints, latent heat decreases with increasing elevation, leading to lower ET in high-Alpine sites, as the higher values of the Bowen ratio for the high-Alpine sites indicates. Phenology is also affected by elevation, the average day of the year when the simulated growing season starts (taken as the mean day when the biomass is higher than the biomass threshold at cut height) increases with increasing elevation. It spans from mid-March in the pre-Alpine site CH-Cha to mid-May in the high-Alpine sites of IT-MBo and IT-Tor. Consequently, the mean yearly GPP is higher where the growing season starts earlier. For instance, the resulting mean annual GPP in CH-Cha is more than the double compared to IT-Tor. However, the average GPP of the month July only, intended as a proxy for the GPP in the maximum growth period, does not differ much across the sites, and lower values are rather indicative of water limitations, testifying similar levels of productivity in the peak of the summer regardless of the site elevation.

The model evaluation against grass biomass dynamics clearly shows that snow presence on the ground limits the growing season at higher altitudes (Figure 4). The model responds to the inter-annual variability of the snow cover. In years with large snow accumulation, the growing season starts later compared to years with a less persistent snow pack. For instance, the years 2011 and 2013 in IT-Tor are characterized by lower and higher than average snow depth respectively. The following growing seasons is respectively anticipated and delayed in both simulations and observations.

In CH-Cha, CH-Fru and CH-Oe1the total simulated harvested biomass falls within, or is very close to, the range reported by observational studies. Considering the large variability in published biomass estimates across sites and even within the same site, it is difficult to conclude if such differences are a model shortcoming or simply dictated by observation uncertainty. In

CH-Oe1 the LAI dynamics are also well captured. The simulated biomass and LAI patterns in DE-Fen, DE-RbW and DE-Gwg fit the more detailed field data for 2015 (Figure S4). Also, the LAI in AT-Neu is simulated well, and the length of growing season and the range of variability of available observations are matched. However, there are discrepancies on the exact dynamics of grass cuts in several sites, most pronounced at AT-Neu. These are expected as grass cuts are prescribed at regular intervals in the model, while they may occur irregularly in reality (for instance dictated by specific weather events) and they might also vary from year to year. The simulation in IT-Tor matches quite well the magnitude of the grass biomass and the beginning of the growing season, but overestimates its length by approximately one month, which explains the major discrepancy between simulated and observed LAI and leaf-biomass.

Simulations of the lysimeter data in DE-Fen, DE-RbW and DE-Gwg provide the opportunity to test the soil biogeochemical dynamics and especially nutrient leaching as well as biomass productivity. The harvested dry matter and, consequently, also harvested nitrogen are considerably underestimated by the model compared to the lysimeters data. The simulated mean annual leaching of DOC in the years 2012-2014 shows values of the same magnitude of observations across all the three sites. The model estimates accurately N- $NO_3^-$ leaching in DE-RbW and DE-Gwg, while it underestimates it in DE-Fen, where observations vary more than for the other sites (Figure 5).

Since the information required to initialize the model carbon and nutrients pools is not available, a-posteriori we compare the C:N values obtained through the spin-up process with the values reported in literature for each site (Table S3). Although measurements might be affected by considerable uncertainty, it is clear that the model underestimate the observed C:N (Table S3). In the sites of DE-Fen and DE-RbW, simulated C:N soil are 5.29 and 5.00 respectively while the observed values are much higher (8.8 and 8.9 respectively). Such discrepancy might be due the combination of the relatively low C:N of litter and the low C:N of manure used in the simulations, which is fixed to 8.9 as suggested by literature (Fu et al., 2017). To ensure that this assumption is not problematic, we perform a series of simulations for the site DE-RbW fertilizing the system with manure characterized by an increasing C:N spanning from 10 to 25, but preserving the absolute amount of N introduced in the ecosystem. As expected, the soil C:N increases with increasing manure C:N (Figure S7a), while the simulated leaching of N and NPP do not remarkably change across the experiments (Figure S7b and Figure S7c),. These results highlight the importance of the site's history for the determination of the exact C:N value, but at the same time they suggest that this is not particularly critical in the model to determine nitrogen leaching and grassland productivity, which are controlled by the actual amount of N, rather than by C:N.

## 3.2 Numerical fertilization experiments

The simulated $NO_3^-$ concentrations in the leaching flux span two orders of magnitude ranging between 0.03 and 12 mg L$^{-1}$ (Figure 6a), compared to an environmental limit of 50 mg L$^{-1}$ in Europe (EEC, 1998). In this case we prefer comparing $NO_3^-$ rather than $NO_3^-$-N concentrations to the widely known threshold of 50 mg $NO_3^-$ L$^{-1}$ imposed by the European Directive. The high-Alpine site IT-Tor exhibits the highest $NO_3^-$ concentration, increasing almost monotonically with increasing nitrogen

input (Figure 6a). The other high-Alpine site IT-MBo shows concentrations comparable to the other sites but with higher interannual variability. Except for IT-Tor, for the other sites, the $NO_3^-$ concentration in leaching does not increase monotonically with the input in the range of N inputs between 30% and 50% of the maximum fertilization (Figure 6a).

The variability of $NO_3^-$ leaching concentration is remarkable not only across sites belonging to different elevation classes, but also within the same elevation range. The largest variability among sites of similar elevation is in the N input range of 30 to 50% of the maximum N fertilization. For example, the difference in $NO_3^-$ leaching concentration between AT-Neu and CH-Fru at 50% of maximum fertilization is comparable to the difference between the leaching obtained in AT-Neu increasing the N-load from 50 to 100%. The same observation applies to the pre-Alpine sites of DE-Fen and CH-Oe1.

Harvested N as a function of the nitrogen fertilization in the high-Alpine sites IT-MBo and IT-Tor differs remarkably from the other sites (Figure 6b). At these high-Alpine sites, the harvested-N does not considerably increase with increasing N inputs, as grass productivity is likely constrained by other factors than N. For the other sites, a clear increase of harvested N emerges for N fertilization spanning from 10% to about 40% of the maximum. For values higher than 40%, there is not much gain in simulated harvested N (and thus biomass) regardless of the increased nutrient availability.

When results are summarized in the harvested N versus N leaching space, this difference is more pronounced (Figure 6c). Patterns of pre-Alpine and Alpine sites partly overlap, but occupy a distinct space away from high-Alpine sites. High-Alpine sites show a limited range of variability of harvested N compared to the range of leaching $NO_3^-$. Beyond a certain threshold of N input also the other pre-Alpine and Alpine sites show limited increase in harvested N. Also note that for lower N-input the $NO_3^-$ leaching is to be relatively stable while harvested N and thus grass biomass increase. The N contributing to grass growth

is one order of magnitude higher than the leaching N in the high-Alpine sites. In comparison, for pre-Alpine and Alpine sites it is two orders of magnitude higher testifying a closer N-cycle despite more intensive fertilization.

When the harvested N and the N concentration in groundwater recharge are combined in a ratio, we obtain an index, which depicts the efficiency of the fertilization practice (Figure 6d). Ideally, this index should be maximized to obtain a win-win situation which optimizes grass yield and minimizes water quality issue. When plotting the index as a function of the

percentage fertilization, the pre-Alpine and Alpine sites exhibit a range where the index is maximum. This range is in between 20% and 60%. Specifically, CH-Cha, CH-Oe1, DE-Fen, DE-Gwg and CH-Fru exhibit the highest value at 20% of the fertilization input (67-100 kg N ha$^{-1}$ yr$^{-1}$), DE-RbW at 35% of the maximum load (175 kg N ha$^{-1}$ yr$^{-1}$) and AT-Neu at 50% of the maximum load (167 kg N ha$^{-1}$ yr$^{-1}$). The high-Alpine sites do not exhibit any optimum, but only a monotonically decreasing line, as N fertilization does not stimulate growth but rather increase $NO_3^-$ leaching.

As a matter of fact, correlating the efficiency index with the elevation (R = -0.66 p =0.05) of each site shows a predominant decreasing trend with increasing elevation (Figure 7a). However, there are exceptions such as CH-Oe1, DE-Gwg and CH-Fru showing low index despite relatively low elevation. The correlation of the index with the fraction of precipitation, which is lost through groundwater recharge, provides an even better descriptor (R = -0.75 p =0.02) of the fertilization efficiency. We

also correlate the index to the soil hydraulic conductivity (Figure S6) but the correlation is not significant ($R=-0.28$ $p=0.47$).

Low water leakage fractions are associated with higher value of the N-efficiency index (CH-Cha, DE-Fen, DE-RbW, AT-Neu) while sites characterized by higher groundwater recharge have a low index (CH-Oe1, DE-Gwg, CH-Fru, IT-MBo, IT-Tor), highlighting a dominant hydrological control beyond soil-element biogeochemistry. In the sites CH-Oe1, CH-Fru, IT-MBo and IT-Tor the high groundwater recharge also corresponds to high soil hydraulic conductivity (Ks), while in DE-Gwg Ks is relatively small (Figure S6). On the contrary, in AT-Neu the groundwater recharge fraction is relatively small and the N-

efficiency index is high despite a high hydraulic conductivity (Figure S6).

### 3.3 Numerical climate change experiments

An increase of air temperature of +3 °C and CO2 of +250ppm result in higher grassland productivity and a change in the water balance with higher ET. Specifically, ET increases on average across all the sites of 88 mm (8.2 %), while water leakage decrease of a corresponding quantity in mm to close the water budget (Figure 8). The enhanced productivity of grasslands

leads to higher yields, which, however, do not increase at the same rate across the different sites. While pre-Alpine and Alpine sites are simulated to experience an average yield increase of 17% compared to the current climate scenario, grasslands in high-Alpine sites are expected to produce up to +73% (IT-MBo) and +120% in (IT-Tor) (Figure 8). The reason for such a variability resides in the different increase of the length of the growing season. With higher temperature the sites IT-MBo and IT-Tor would benefit of 25 and 45 days longer growing season respectively, while the other sites would experience a 13-days

longer growing season on average. Despite such a high relative increase, the absolute yields in high-Alpine sites would still be remarkably lower than in pre-Alpine and Alpine sites (Figure S5a). For the same N input, the N losses to the environment through leaching are expected to decrease for modified climate conditions compared to the historical climate (Figure 8).

As a result, the efficiency index generally increases, highlighting a more efficient use of N (Figure S5b) and in some cases (i.e., DE-Fen, DE-Gwg, CH-Fru) the optimal N input may also increase. The efficiency index still shows a predominant

decreasing trend with increasing elevation (Figure 7a), although the correlation is weaker ($R = -0.53$ $p = 0.14$) compared to the historical climate, while the fraction of precipitation which is lost through groundwater recharge is still confirmed to be a very good descriptor ($R = -0.74$, $p = 0.02$) of the fertilization efficiency (Figure 7b).

## 4 Discussion

### 4.1 Fully-integrated mechanistic ecosystem modelling: successes and limitations

While many ecosystem modeling applications have been discussed in literature, including detailed ecohydrological (e.g., Ivanov et al., 2008; Tague et al., 2013; Millar et al., 2017) and soil biogeochemistry applications (e.g., Parton et al., 1998; Kraus et al., 2014; Robertson et al., 2019), rarely, if ever, a single integrated model has been tested across different compartments and disciplines in concurrently reproducing surface energy budget, hydrological dynamics, vegetation productivity, and nitrogen budget. Here, we raise the bar to challenge T&C-BG, in reproducing these processes across nine

grassland sites in the broad Alpine region. Furthermore, to ensure future model transferability and avoid local tuning, we also use the same vegetation and soil biogeochemistry parameters across all sites, with few exceptions where an elevation or latitudinal dependence of a parameter should be preserved. Despite such an "average" parameterization, the model responds surprisingly well to the challenge as energy and carbon fluxes, soil hydrology, vegetation dynamics, and $NO_3^-$ and DOC leaching fluxes are all with realistic magnitude and similar to observations with few notable exceptions discussed below. Also

feedbacks between compartments are realistic in the model, as it is the case of growing season length varying depending on the date of complete snow cover disappearance or the limitations in grass growth and thus LAI at low nitrogen availability (Figure 6a). Overall, simulations suggest that a correct representation of phenology and thus length of the growing season is a fundamental aspect of model performance as peak of the season GPP is much more similar across sites than annual GPP values (Table 4).

Despite such a positive outcome, there are a number of uncertainties in both the observations and model simulation that are relevant to highlight. It is well-known that flux towers do not close the energy budget (e.g., Foken, 1998, 2008; Wilson et al., 2002; Widmoser and Wohlfahrt, 2018; Mauder et al., 2020). The model generally overestimates the sensible heat compared to observations, thus suggesting that the missing energy is most likely attributable to sensible heat as supported by other studies (Mauder et al., 2006; Wohlfahrt et al., 2010; Liu et al., 2011) and justifies the lower $R^2$ for sensible heat compared to the other

fluxes. Observations of soil water content depend on the specific soil hydraulic properties and microtopography in the location where the sensor is installed and are also often subject to temporal drifts (Takruri et al., 2011; Mittelbach et al., 2012). The model represents a vertically explicit but spatially implicit average soil moisture over the tower footprint. For this reason, even though we normalize soil moisture using effective saturation, the comparison should be seen more in qualitative terms rather than attempting to reproduce exactly the observed soil moisture values. Most important, in terms of vegetation productivity,

biomass data reported from different articles present a remarkable variability despite referring to the same study site. This might depend on differences in sampling protocol and instrumentation (Zeeman et al., 2019) as well as natural spatial variability. In light of these uncertainties we do not dwell in explaining model to data differences as far as the long-term magnitude of observed biomass is similar. The only exception is the very significant underestimation of the simulated biomass compared to lysimeters measurements. We attribute large portion of this inconsistency to the well-documented lysimeter

oasis/border effect, which generates crop yields and evapotranspiration fluxes 10-20% larger than larger-field observations (Oberholzer et al., 2017). As the model captures well the pattern of biomass and carbon fluxes in the flux-tower locations, it would be difficult to justify why biomass productivity should be much different a few hundreds of meter apart in the lysimeters under similar management (Fu et al. 2017). The model does not properly represent the inter-annual variability of harvested carbon and N, for example not capturing the higher productivity of 2013, a particularly productive year due to the reduced

snow cover (Zeeman et al., 2017). One explanation can be that we input to the model the same management strategy every year while local management vary from year to year (see SI in Fu et al., 2019). Inconsistencies on the number of manure applications or grass cuts as well as a discrepancy of the order of weeks on the day of the actual manure application or grass

cut might occur in the simulations in a given year. More generally, simulation results can be affected by the spin-up process performed to initialize the carbon and nutrient pools in the system in absence of historical information. The induced stationarity might influence the soil organic carbon and nitrogen pools enrichment or depletion and can generate discrepancies with observations. For instance, simulated C:N values are considerably lower than values reported in literature for each site (Table S3), likely due to the quite low value of C:N used for the manure stoichiometry (Fu et al., 2017), which carbon is in part quickly respired, as manure is modeled similarly to fresh litter. However, the absolute value of added nitrogen in the soil is much more important than the C:N ratio itself in controlling N leaching and grassland productivity (Figure S7), attenuating this problem in the computation of the nitrogen fluxes. Other possible N losses in the environment such as ammonia volatilization or denitrification ($N_2$) do not affect the N-budget significantly, being their average losses across the three German sites in the order of 0.74 kgN ha$^{-1}$ yr$^{-1}$ and 0.28 kgN ha$^{-1}$ yr$^{-1}$ respectively.

Simulated $NO_3^-$ concentration in groundwater recharge in DE-Fen is underestimated compared to the observed values from lysimeters. While this can be likely due to model shortcoming, the model uncertainty is not the only factor contributing to this incongruence as also the discrepancy of management between the model and reality might play a role. Moreover, the model does not take into account preferential flows, which might be particularly pronounced in the lysimeters, (Schoen et al., 1999; Groh et al., 2015; Benettin et al., 2019; Shajari et al., 2019). Regardless of these existing differences, and keeping in mind that the exact value of the results is likely more uncertain that the comparison across scenarios/sites, we argue that results obtained with T&C-BG are more encouraging than discouraging and allowed us to explore the complex functioning of the overall ecosystem and to carry out virtual numerical experiments that can inform farmers and legislators.

## 4.2 Simulated responses to fertilization and climate change: management implications

A term of comparison for the simulated $NO_3^-$ leaching concentrations is the limit of 50 mg $NO_3^-$/L imposed on groundwater nitrate concentrations by the European Drinking Water Directive (98/83/EC, EEC, 1998). Results from the numerical experiments under historical climatic conditions show that $NO_3^-$ leaching concentrations are generally lower or much lower than this threshold, even under highly intensive management practices. A scenario with temperature increase is expected to lower such leaching concentrations in favor of higher biomass productivity (Figure 8 and Figure S5a).

The EU Nitrate Directive 91/676/EEC (EEC, 1991) imposes the maximum N fertilization rate for stable managed grasslands of 170 kgN ha$^{-1}$yr$^{-1}$. The Directive leaves room for flexibility to European countries, since they are allowed to introduce higher N fertilization loads if they demonstrate that grassland absorbs it efficiently. However, if the monitoring of surface and groundwater $NO_3^-$ concentration reveal water quality issues, then EU can intervene with an infringement proceeding. As an example, Germany in 2017 re-adapted the national Fertilizer Ordinance of 2007 after an infringement proceeding by the EU (Kuhn, 2017).

The threshold of 170 kgN ha$^{-1}$yr$^{-1}$ is computed on the basis of a NO$_3^-$ input-output balance at the farm scale. In such a simple computation the NO$_3^-$ losses in the environment are assumed to be 30% of the input (IPCC, 2000, 2006). These losses include

both groundwater recharge and surface runoff. Although some countries have lowered this threshold to 10%, such as Ireland, or to 20%, such as Switzerland, there is evidence that the threshold is quite high compared to observed losses in grasslands (Eder et al., 2015). Our study confirms such a finding highlighting NO$_3^-$ concentrations in the order of 0.3-12 mg L$^{-1}$ for historical climate conditions, although we only consider the NO$_3^-$ losses with groundwater recharge and neglect those through surface runoff.. However, there is evidence that NO$_3^-$ losses through surface runoff are usually small compared to losses

through groundwater recharge ( Jackson et al., 1973; Casson et al., 2008). Despite the noticeable inter-site variability, the simulated losses of NO$_3^-$ into groundwater are generally lower than 10% of the N-input, even under highly intensive fertilization scenarios (Figure 6c). As a result of increasing temperature, this percentage is expected to decrease even further, due to longer growing seasons and higher ET, which lead to a more efficient use of the N input (Figure 8).

Among the N inputs explored in the analysis, the value that guarantees the maximization of the N-efficiency ratio, for the

current climate, is quite close or lower than the limit of 170 kgN ha$^{-1}$ yr$^{-1}$ in pre-Alpine and Alpine sites (Figure 6d). Thus, we confirm that such a limit is a reasonable upper threshold. However, an emerging aspect from the simulations is the large variability across sites in NO$_3^-$ leaching and response to fertilization. While model uncertainties exist and we are unlikely to capture the exact magnitude of all N-fluxes at all sites, this variability is likely underestimated rather than overestimated by the model due to simplifying assumptions and commonality of parameters. The NO$_3^-$ leaching and the grassland yields depend

on the degree at which nitrogen is limiting productivity versus other environmental factors (e.g., temperature), on the capability of vegetation in taking up nutrients, and on the hydrological characteristics of the sites. A soil favoring water drainage or a fast snowmelt at the end of the winter generates larger amount of N leaching into groundwater. Most likely one factor does not exclude the other (Figure 7 and Figure S6), but the combination of both soil properties and hydrological regime drives the correlation between N leaching concentration and groundwater recharge fraction. The N uptake efficiency of grassland depends

more on the nutrient demand with a longer growing season (Zeeman et al. 2017) higher temperatures and lack of water stress favoring the N demand, even though the importance of the different aspects is difficult to disentangle (Lü et al., 2014). In the simulations, the length of the growing season at CH-Oe1, DE-Gwg and CH-Fru is comparable to the other pre-Alpine and Alpine sites (Table 4). However, simulated NO$_3^-$ leaching is higher and the produced biomass is lower. This is especially true at low N fertilization levels. This further remarks the important role of local hydrology (Fig 6a-c) on N-cycle. The comparison

between current and modified climate scenarios further remarks the importance of the interplay among various factors, since higher temperatures increases the efficiency in using N, due to longer growing season and consequent higher N uptake. In the simulations, an increase of air temperature has a positive impact on the overall grassland functioning, however, multi-decadal effects should be evaluated more carefully, since long-term impacts on soil biogeochemistry – not analyzed here- might affect the fertility of the site (Schlingmann et al., 2020). Literature shows that also the richness of species composing the grassland

might favor nutrient uptake (Tilman et al., 1996; Spehn et al., 2005; Niklaus et al., 2006), but this type of ecological feedback is not implemented in the model, which simply considers an "average species composition" as a representative grassland. While grass composition and botanical heterogeneity might be important for the ecological functioning of the system, they are unlikely to affect significantly carbon and water fluxes at the ecosystem scale, at least to a level that goes beyond current model and observational uncertainties as demonstrated by the match between simulations with common parameters and flux-tower observations.

As local hydrological and soil conditions (and potentially also different site biogeochemical history and species composition– not considered here) modify the grass response to N fertilization, disparities among farmers might emerge simply as a function of location. Farmers owing a field in an area characterized by higher groundwater recharge might be disadvantaged because losses of $NO_3^-$ to groundwater could be higher and yields limited for the same amount of N fertilization. On the contrary, farmers owing fields in hydrological favorable sites might be advantaged. Such disparities should be taken into account in two ways: first in delineating location specific fertilization limits and second in applying some sort of compensation for locations/regions that are disadvantaged. A good practice in this direction is the proof of ecological performance (PEP), introduced in Switzerland (Swiss Federal Council, 1998). Direct payments are transferred to farmers who join this program, which imposes a wealth of rules in order to preserve the environment. Among the requirements the computation of the farm $NO_3^-$ balance needs to be also provided.

On the basis of modeling results, regulations based on a fixed threshold for the whole European grasslands could be largely suboptimal. Even if we might underestimate losses because of model uncertainties and because we do not consider surface runoff, resulting $NO_3^-$ losses are abundantly lower than 30% of the N-input and can even decrease with a warming climate. Overall, there is still leverage for enhancing N fertilization before reaching the threshold assumed by the EU Directive. However, we also showed that higher N-load do not necessarily correspond to higher yields, especially above certain thresholds. Therefore, even though $NO_3^-$ leaching is low, an effective increase of manure application might not be beneficial for grass yield and should be evaluated on the basis of the site-specific characteristics. Fixed threshold based regulations might admit more argument in view of climate change, since response to warming might widely differ across Alpine regions.

We suggest as a possible alternative and better strategy the definition of thresholds on fertilizer input based on a distributed mapping of the landscape. In absence of more precise information, agricultural areas could be classified based on the crop $NO_3^-$ use efficiency, soil type, and hydrological characteristics. For instance, Klammler et al. (2013) provide and interesting example of $NO_3^-$ leaching mapping in a case study in Austria. Alternatively, mechanistic ecosystem models as used in this study could be employed to identify ranges of optimal fertilization levels, here identified as the maximum in the  ratio between harvested N and N leachate (Figure 6d), for different sites, groups of sites, or even in a fully distributed manner. This process could be demanding in terms of resources as requires advanced expertise, sufficient data to constrain the models, and adequate modelling tools (Decrem et al., 2007) . However, with increasing computational capabilities and data availability to constrain

model parameters, it is likely that mechanistic modeling approaches might become more popular and an essential tool for fostering the mapping process and provide distributed information to refine environmental regulations.

**5 Conclusion**

We simulated the dynamics of nine managed grassland sites across the broad European Alpine region. We applied the mechanistic model T&C-BG which fully integrates land-surface mass and energy fluxes, soil hydrology, vegetation dynamics, and soil biogeochemistry. The model was confirmed to reproduce realistic magnitude and temporal dynamics of ecohydrological variables across multiple compartments in an effort of model evaluation that goes beyond many of the existing attempts to confirm ecosystem models. The model was subsequently used to quantify the impact of different N fertilization

scenarios, with focus on grass productivity and $NO_3^-$ concentration in groundwater recharge. Each management scenario was simulated under current and modified climatic conditions (i.e., +250 ppm of atmospheric $CO_2$ concentration and +3 °C in air temperature). Simulations reveal that although groundwater recharge concentrations are relatively small and well below environmental policy limits, there is high variability across grasslands, also for sites located at similar elevation. Such variability is mainly driven by local environmental controls on productivity that reflects in the grass capability to take up

nutrients and by the differences in the hydrological regime summarized as the fraction of precipitation that becomes recharge. Under modified climate scenarios, the variability across sites remains and increased temperature results in higher N use efficiency, especially in high-Alpine sites, at least for the timeframe analyzed here. We suggest that these factors should be taken into account by legislators while defining thresholds on fertilizer loads. Guidelines based on the site-specific rather than based on fixed thresholds across large regions would favor the maximization of grass yields while allowing preserving similar

water quality targets. Fully-integrated mechanistic ecosystem models as employed here have a big potential as tools to construct these maps under current and future climate scenarios.

**Code availability**

The model code is available at the link: https://github.com/simonefatichi/TeC_Source_Code (last access December 2020).

**Data availability**

FLUXNET data for running and confirming T&C-BG plot-scale simulations were downloaded from https://fluxnet.fluxdata.org/. Specifically we downloaded data for the stations Chamau (DOI: 10.18140/FLX/1440131), Oensingen (DOI: 10.18140/FLX/1440135), Früebüel (DOI: 10.18140/FLX/1440133), Neustift (DOI: 10.18140/FLX/1440121), Monte Bondone (DOI: 10.18140/FLX/1440170) and Torgnon (DOI: 10.18140/FLX/1440237). Data

for the sites Fendt, Rottenbuch and Graswang are available in Zenodo, divided into management and plant physiology data (DOI: 10.5281/zenodo.4267810) and meteorology, environment and surface flux data (DOI: 10.5281/zeonodo.427887).

## Author contribution

MB and SF conceived the ideas and discussions with MZ supported their development. MB and SF designed the experiments, MB ran simulations and developed the analyses. All authors contributed to the writing.

## Competing interests

The authors declare that they have no conflict of interest.

## Acknowledgements

This study was supported by the DAFNE project (https://dafne.ethz.ch/), funded by the Horizon 2020 programme WATER 2020 of the European Union (grant agreement no. 690268).

The authors acknowledge Dr. Gianluca Filippa from the Environmental Protection Agency of Aosta Valley for sharing data of the site IT-Tor. A special acknowledgment to Dr. Ralf Kiese from KIT-IfU for the insightful comments to the manuscript. We also acknowledge Dr. Edoardo Cremonese and Dr. Stefano Manzoni for the extremely helpful reviews of the original version of this manuscript.

This study built upon TERENO/ScaleX cooperation and we thank Benjamin Wolf, Nadine Ruehr, Heather Shupe, Martina Bauerfeind, Carsten Malchow, Maximilian Graf, Matthias Mauder (all KIT/IMK-IFU) and the Scientific Team of ScaleX Campaign 2015 for their contribution. The TERrestrial Environmental Observatory (TERENO) infrastructure is funded by the ATMO program of the Helmholtz Association and the Federal Ministry of Education and Research. MZ received support from the German Research Foundation (DFG; project ZE 1006/2-1).

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

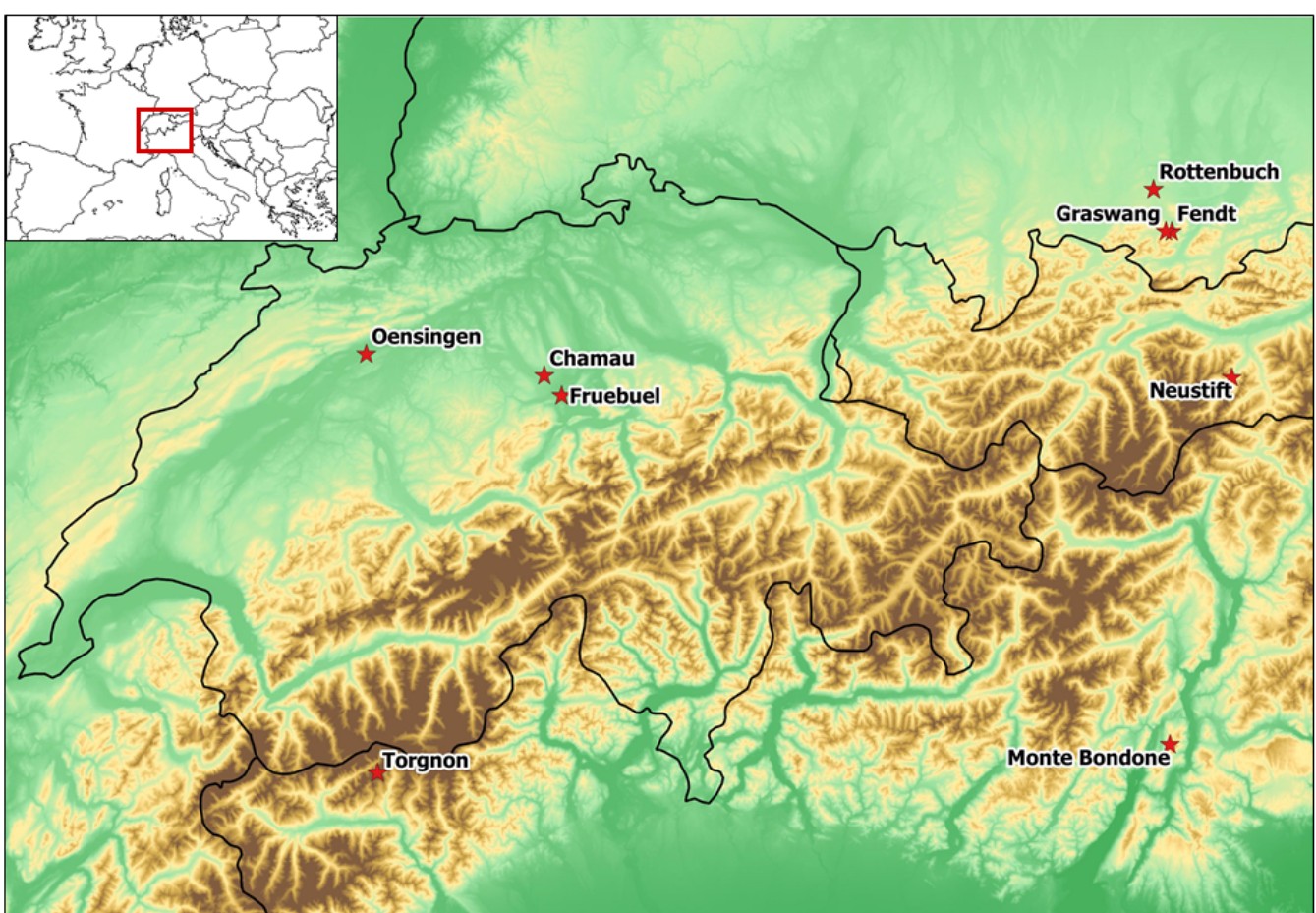

**Figure 1. Location of the study sites.** The 9 sites are located across the European Alps in Italy (IT-Tor, IT-MBo), Switzerland 940   (CH-Cha, CH-Oe1, CH-Fru), Austria (AT-Neu) and Germany (DE-Fen, DE-RbW, DE-Gwg). The map was produced by the Authors and the map with countries' borders was retrieved from "Made with Natural Earth" and the DTM from SwissTopo.ch.

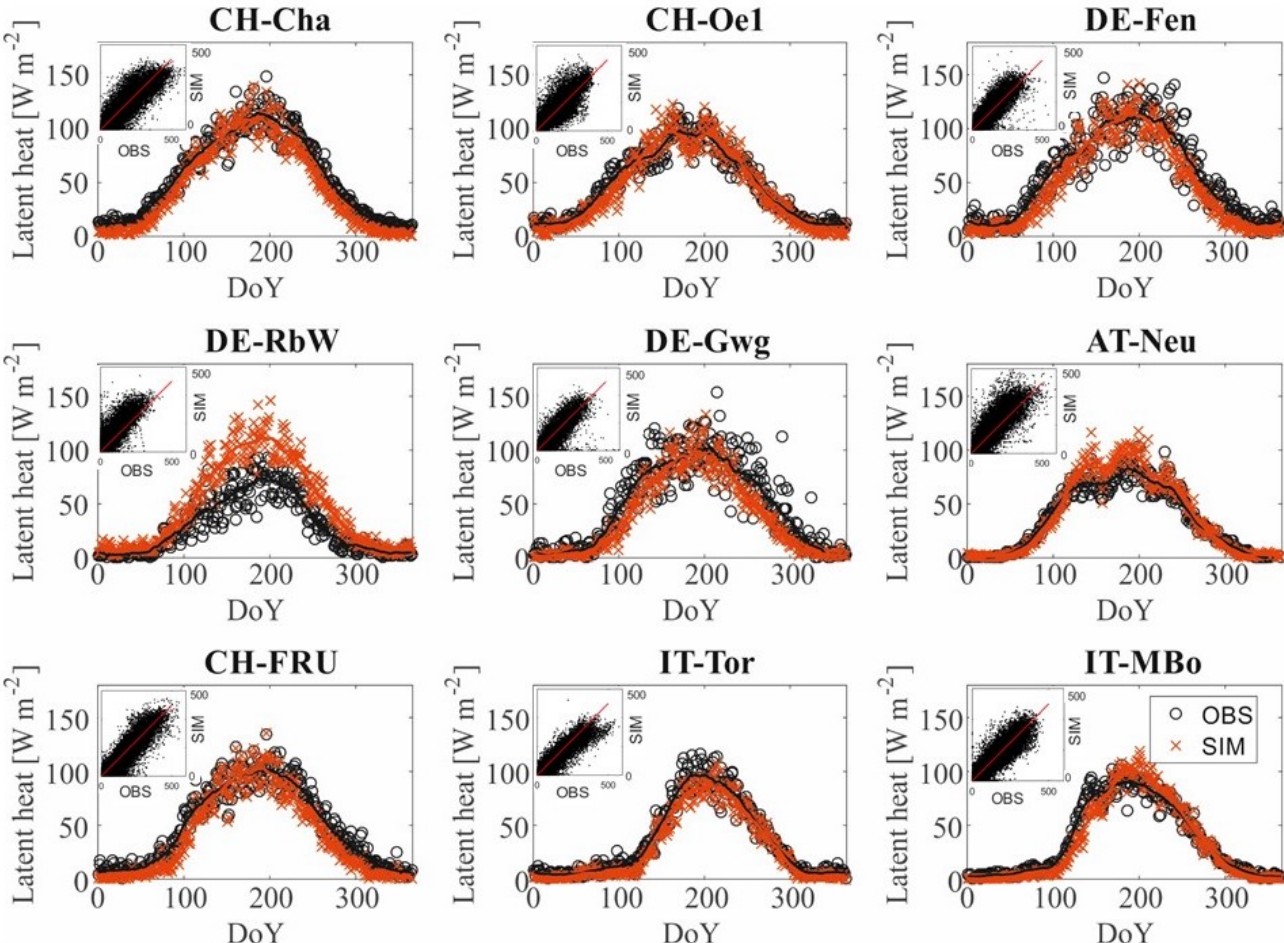

**Figure 2. Observed versus simulated seasonal daily latent heat fluxes.** We compare the observed (black circles) and simulated (red crosses) seasonal pattern of latent heat computing the average value for every day of the year (DoY) considering all the years for which observations are available. We also apply a moving average with a centered window of 30 days (continuous lines). In the upper left corner of each subplot a scatter plot comparison of the hourly values of observed and simulated latent heat is shown. In red is plotted the 1:1 line.

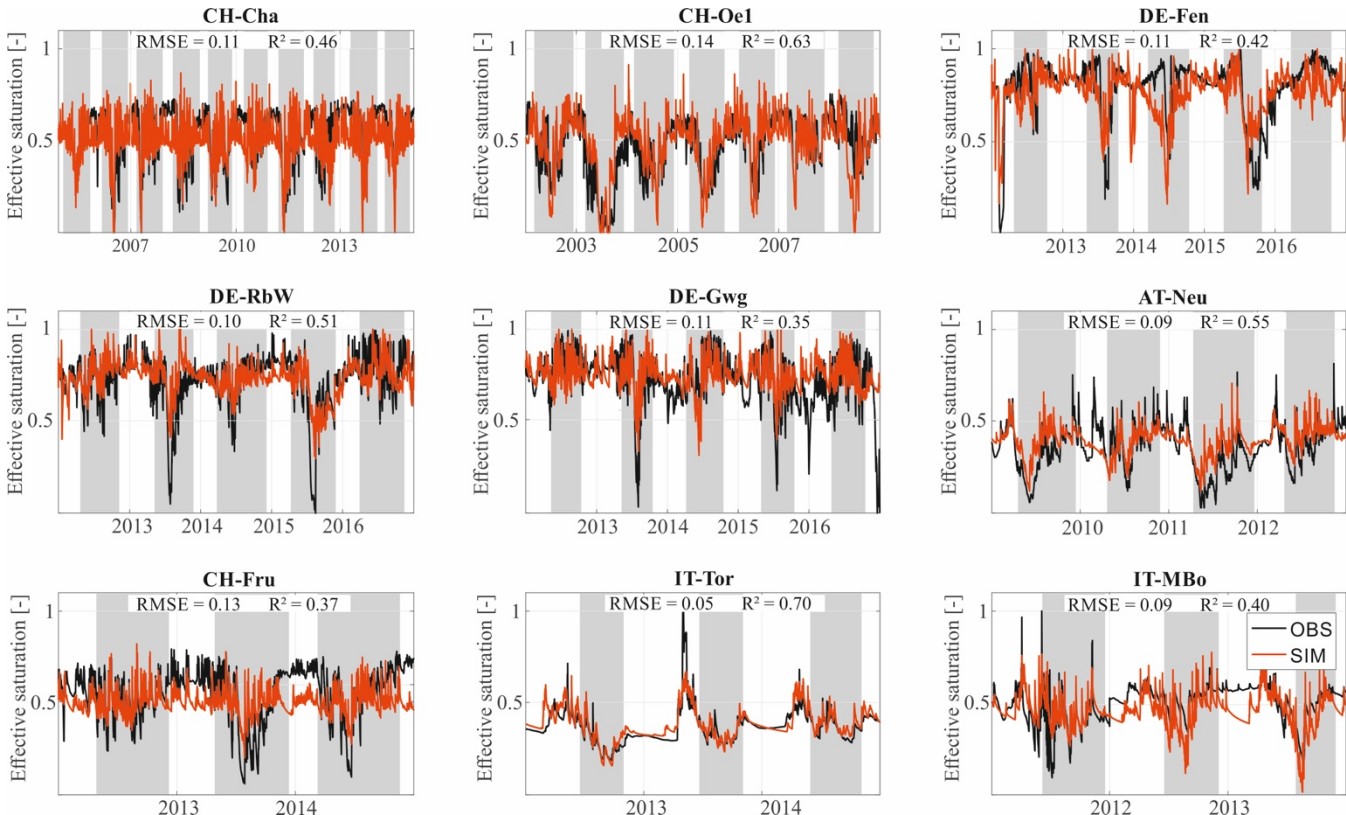

**Figure 3. Observed vs simulated daily effective saturation.** We compare the observed (black) and measured (red) pattern of the effective saturation across all the sites. In CH-Cha, AT-Neu, CH-Fru, IT-Tor soil water content is measured at 5 cm depth, in CH-Oe1 and IT-MBo at 10 cm depth and in DE-Fen, DE-RbW and DE-Gwg at 12 cm depth. The goodness of fit metrics Room Mean Square Error (RMSE) and coefficient of determination ($R^2$) are reported for each station in each subplot. Gray areas represent the growing seasons for each year and site.

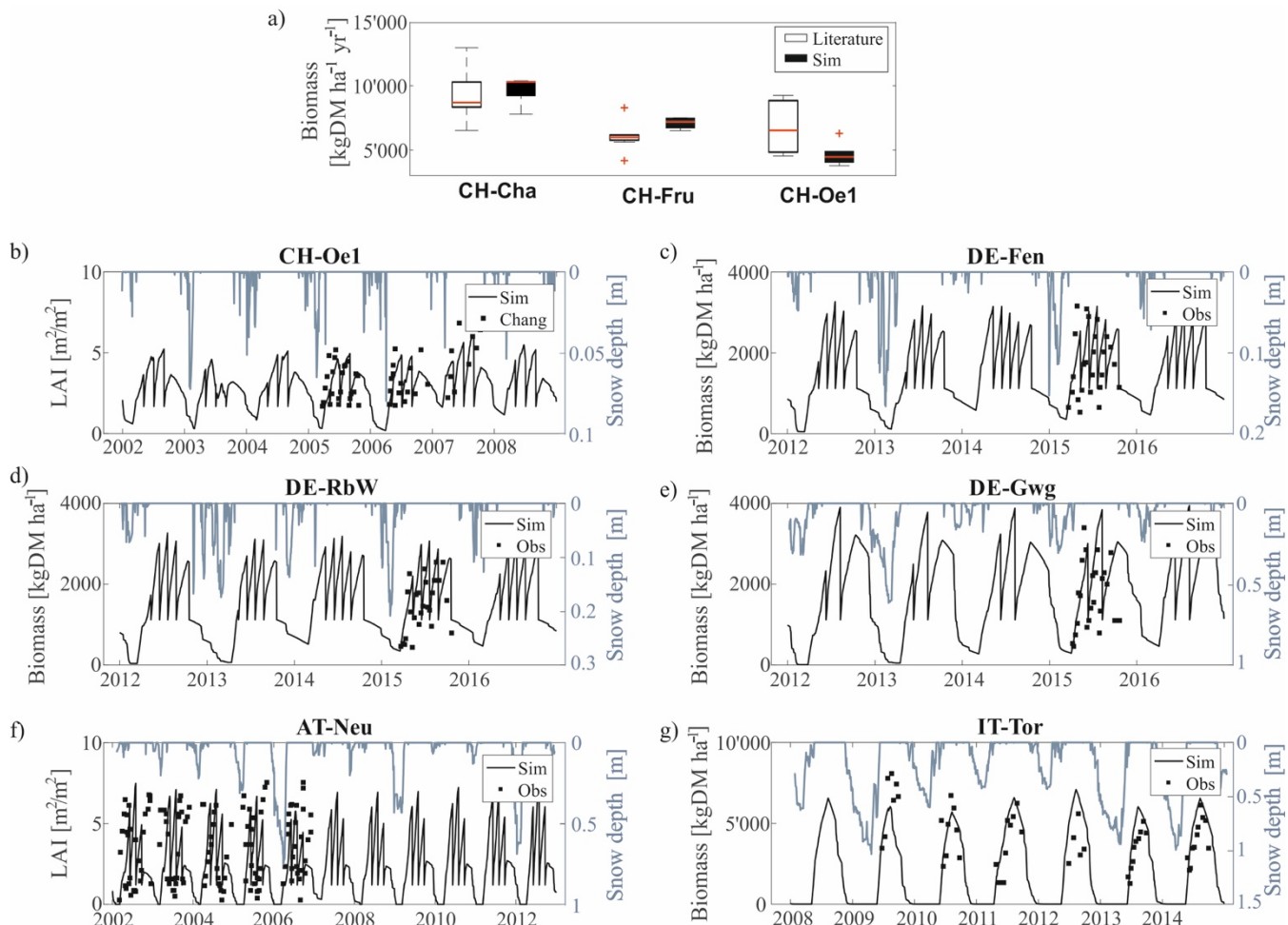


**Figure 4. Observed versus simulated leaf-biomass and LAI.** (a) The simulated mean yearly harvested biomass in CH-Cha, CH-Fru and CH-Oe1 are compared with published values. For CH-Cha data are extracted from Gilgen and Buchmann (2009), Zeeman et al. (2010) and Prechsl et al. (2015). For CH-Fru we compare with data from Gilgen and Buchmann (2009) and Zeeman et al. (2010). For the site CH-Oe1 we compare simulated values with data reported by Ammann et al. (2009). (b) LAI

in CH-Oe1 is compared with observations from Chang et al. (2013). (c)(d)(e) Biomass data for the sites in Germany DE-Fen, DE-RbW and DE-Gwg were provided by the ScaleX campaign 2015 (Zeeman et al., 2019). (f) LAI data in AT-Neu were digitalized from Wohlfahrt et al., 2008b (g) biomass data in IT-Tor, observations were provided by the Environmental Protection Agency of Aosta Valley (Filippa et al., 2015). In all the subplots, we compare the simulated either biomass or LAI (black line) with observations (black dots), simulated snow depth (grey line) is also shown.

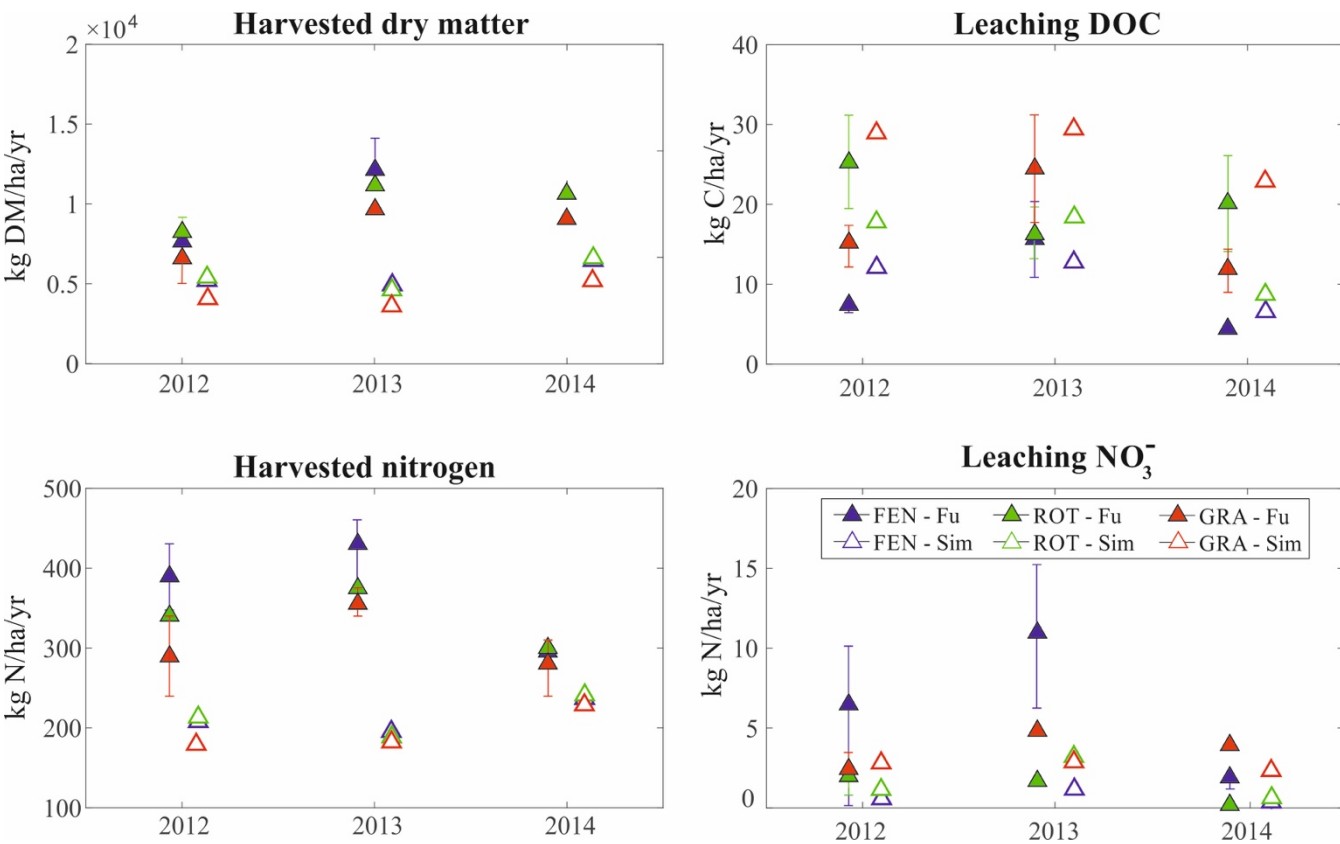


**Figure 5. Biogeochemistry module evaluation.** Observed versus simulated (a) harvested dry matter, (b) harvested nitrogen, leaching of (c) DOC and (d) $NO_3^-$. Simulated annual totals (empty triangles) are compared with annual totals (full triangles) reported by Fu et al. (2017; 2019) in DE-Fen (blue), DE-RbW (green) and DE-Gwg (red). The uncertainty bars represent the 25th and 75th percentile of observations.

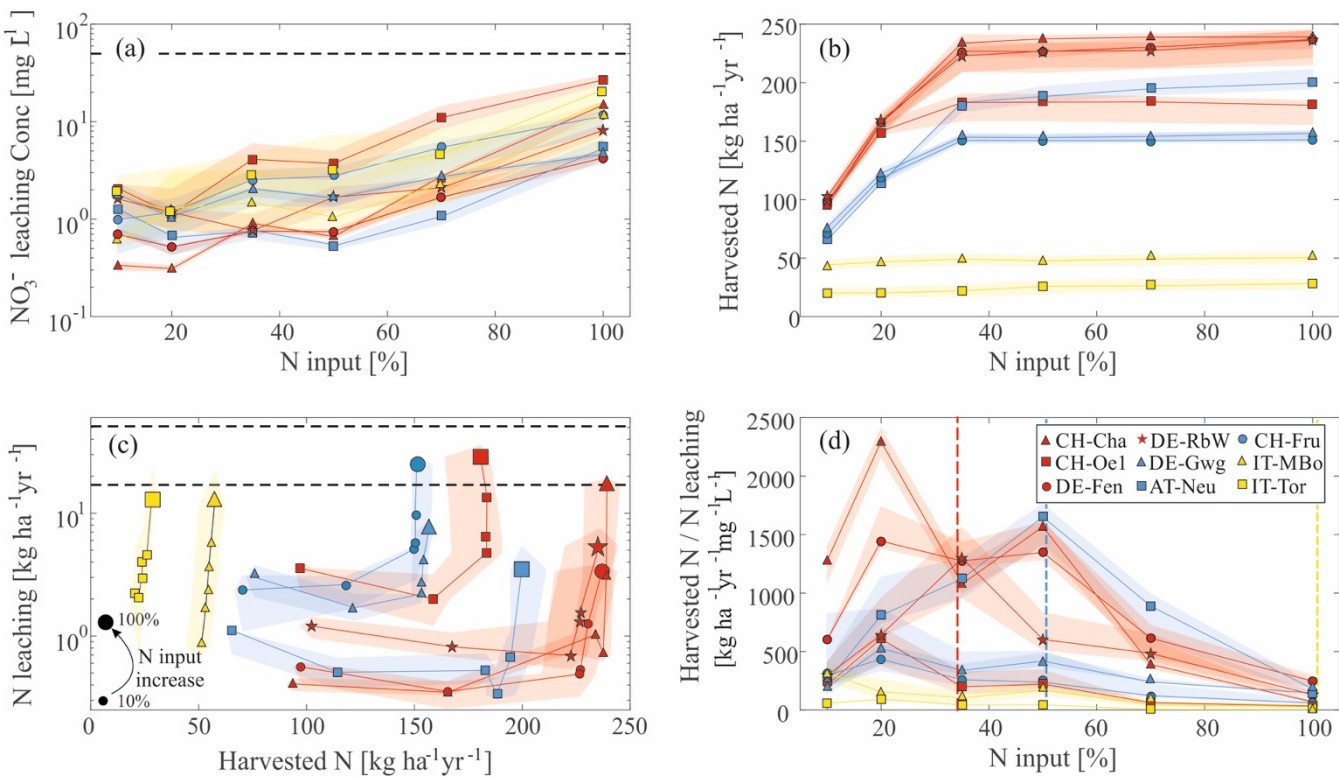


**Figure 6. Results of the fertilization experiments.** (a) $NO_3^-$ concentration in groundwater recharge as a function of different nitrogen fertilization scenarios (percentage of the maximum) in each site. (b) Harvested N in each site as function of different nitrogen fertilization scenarios. (c) Harvested N vs N leaching in each site. Moving counterclockwise follows the increase in N input. As a reference the biggest marker indicates N input of 100%. The dotted lines represent the estimate of maximum

$NO_3^-$ losses assumed by EU regulations or nearby countries. They correspond to the values 17 and 51 kgN ha$^{-1}$ yr$^{-1}$, i.e., 10% and 30% of the maximum allowed input 170 kgN ha$^{-1}$ yr$^{-1}$. (d) N fertilization efficiency index computed as the ratio between the harvested N and N concentration in groundwater recharge as a function of nitrogen input. In all the subplots the colors represent the elevation class, i.e., pre-Alpine (red), Alpine (blue) and high-Alpine (yellow) sites. The colored area around the markers and lines represent the 25$^{th}$ and 75$^{th}$ percentile of the interannual variability of the simulated variables, while the

colored lines connecting data points represent the median values. The vertical dashed bars represent the limit of 170 kgN ha$^{-1}$ yr$^{-1}$ imposed by the EU Nitrate Directive in each of the three classes.

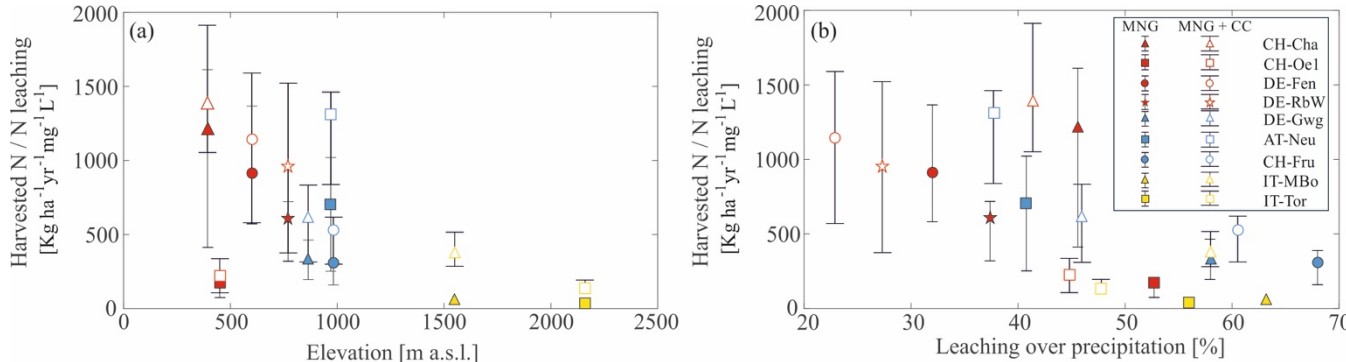

**Figure 7. Ratio between harvested N and N leaching concentration as a function of site characteristics.** (a) Harvested N/ N leaching concentration as a function of elevation in each site. (a) Harvested N/ N leaching concentration as a function of the percentage of yearly water recharge to groundwater over the yearly precipitation. In both plots the whiskers span the 25$^{th}$ and 75$^{th}$ percentile of the interannual variability. Full markers refer to management scenarios (MNG in the legend) with the historical climate, while empty markers refer to management + climate change (MNG + CC in the legend) scenarios.

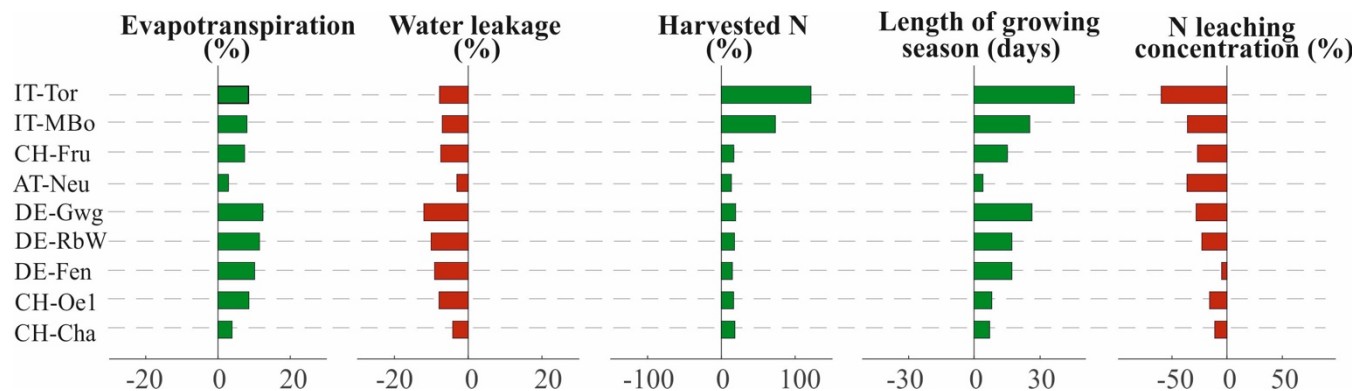

**Figure 8. Changes in evapotranspiration, water leakage, harvested N, length of the growing season, and N concentration in leaching under the influence of modified climate (+3°C and + 250 ppm).** The bar plots indicate the average percentage change of the management scenarios when run under historical and modified climate scenario. The variation in the length of growing season is expressed in days. Red bars refer to a decrease of the variable under modified climate, while green bars represent an increase.

**Table 1. Site characteristics.** The following main sites characteristics are reported: elevation, latitude, longitude, period for which measurements are available, Mean Annual Precipitation (MAP), mean air temperature (mean Ta), number of grass cuts per year and main references to original studies referring to the study sites. The values of MAP and mean Ta are computed from the available time series.

| Site | Elevation (m a.s.l.) | Latitude | Longitude | Years of available data | MAP (mm) | Mean Ta (°C) | # cuts year$^{-1}$ | Soil sand and clay content | Dominant species | References |
|------|------|------|------|------|------|------|------|------|------|------|

| Site | | | | | | | | | | |
|------|---|---|---|---|---|---|---|---|---|---|
| **CH-Cha** | 393 | 47°12' | 8°24' | 2006 - 2014 | 1'134 | 9.4 | 6 | 24% clay 25% sand | Lolium multifl., Trifolium repens | Gilgen and Buchmann, 2009; Zeeman et al., 2010 |
| **CH-Oe1** | 450 | 47°17' | 7°44' | 2002 - 2008 | 1'222 | 9.2 | 3 | 20% clay 60% sand | Lolium perenne, Alopecurus prat., Trifolium repens | Ammann et al., 2007; 2009 |
| **DE-Fen** | 600 | 47°57' | 11°06' | 2012 - 2016 | 850 | 7.8 | 5 | 30% clay 27% sand | Lolium multiflorum Lam., Trifolium repens L. | Kiese et al., 2018 |
| **DE-RbW** | 770 | 47°70' | 10°98' | 2012 - 2016 | 955 | 7.4 | 5 | 29% clay 26% sand | Lolium multiflorum Lam., Trifolium repens L. | Kiese et al., 2018 |
| **DE-Gwg** | 864 | 47°57' | 11°03' | 2012 - 2016 | 1'114 | 5.3 | 2 | 52% clay 9% sand | Lolium multiflorum Lam., Trifolium repens L. | Kiese et al., 2018 |
| **AT-Neu** | 970 | 47°12' | 11°32' | 2002 - 2012 | 856 | 6.8 | 3 | 11% clay 42% sand | Dactylis glomerata, Festuca pratensis, Arrhenatherum elatius, Trifolium sp., Taraxacum officinalis and Ranunculus acris | Hammerle et al., 2008; Wohlfahrt et al., 2008a, 2010 |
| **CH-Fru** | 982 | 47°60' | 8°32' | 2005 - 2014 | 1'627 | 7.6 | 4 | 22% clay 29% sand | Alopecurus prat., Dactylis glomerata, Taraxacum offic., Ranunculus sp., Trifoliu repens | Gilgen and Buchmann, 2009; Zeeman et al., 2010 |
| **IT-MBo** | 1550 | 46°00' | 11°02' | 2003 - 2013 | 1'268 | 5.2 | 1 | 22% clay 29% sand | Nardetum alpigenum | Gilmanov et al., 2007; Vescovo and Gianelle, 2008; Gianelle et al., 2009; Marcolla et al., 2011 |
| **IT-Tor** | 2160 | 45°50' | 7°34' | 2008 - 2015 | 870 | 2.9 | 0 | 10% clay 45% sand | Nardus stricta L., Festuca nigrescens All., Arnica montana L. | Migliavacca et al., 2011; Galvagno et al., 2013 Filippa et al., 2015 |

**Table 2. Numerical experiments.** The sites are divided into pre-Alpine, Alpine, and high-Alpine sites based on the elevation in which they are located. For each class the number of cuts and manure applications as well as the yearly N-load injected in the different modelling experiments is reported. The specific manure application is the same for each site class, but the total amount varies across classes as the number of yearly applications is different. The results are presented as a function of the percentage of max N input, reported in the table for each experimental setup.

| Pre-Alpine/Intensive | Alpine/Extensive | High-Alpine/Extensive | | |
|---|---|---|---|---|
| CH-Cha, CH-Oe1, DE-Fen, DE-RbW | DE-Gwg, AT-Neu, CH-Fru | IT-MBo, IT-Tor | | |
| 5 cuts, 6 fertilizations | 3 cuts, 4 fertilizations | 1 cut, 2 fertilizations | | |
| Yearly N load (kg N ha$^{-1}$ yr$^{-1}$) | Yearly N load (kg N ha$^{-1}$ yr$^{-1}$) | Yearly N load (kg N ha$^{-1}$ yr$^{-1}$) | Specific manure application (kgC ha$^{-1}$ yr$^{-1}$) | % of max N input |

| | | | | |
|---|---|---|---|---|
| 50 | 33 | 17 | 74 | 10% |
| 100 | 67 | 33 | 148 | 20% |
| 175 | 117 | 58 | 260 | 35% |
| 250 | 167 | 83 | 371 | 50% |
| 350 | 233 | 117 | 519 | 70% |
| 500 | 333 | 167 | 742 | 100% |

**Table 3. Coefficient of determination ($R^2$) of the simulated vs observed energy and carbon fluxes from flux towers data.**

| | Net radiation | Sensible heat | Latent heat | GPP |
|---|---|---|---|---|
| CH-Cha | 0.98 | 0.63 | 0.86 | 0.78 |
| CH-Oe1 | 0.98 | 0.66 | 0.82 | 0.69 |
| DE-Fen | 0.97 | 0.51 | 0.79 | 0.66 |
| DE-RbW | 0.74 | 0.48 | 0.76 | 0.63 |
| DE-Gwg | 0.74 | 0.39 | 0.70 | 0.59 |
| AT-Neu | 0.95 | 0.53 | 0.85 | 0.83 |
| CH-Fru | 0.96 | 0.72 | 0.87 | 0.85 |
| IT-MBo | 0.93 | 0.75 | 0.88 | 0.79 |
| IT-Tor | 0.94 | 0.69 | 0.87 | 0.72 |

**Table 4. Simulated energy, water and carbon fluxes at each site.** The mean annual evapotranspiration (ET), mean annual net radiation (Rn), mean annual Bowen ratio (Bo), mean annual gross primary production (GPP), mean gross primary production in the month of July (July GPP) and the mean day of the year in which the growing season starts (Start growing season) are reported. The values are reported as mean ± standard deviation of the interannual variability.

| | ET (mm yr$^{-1}$) | Rn (W m$^{-2}$) | Bo (/) | GPP (gC m$^{-2}$ yr$^{-1}$) | July GPP (gC m$^{-2}$ month$^{-1}$) | Start growing season (DoY) |
|---|---|---|---|---|---|---|
| CH-Cha | 599±35 | 67.7±3.8 | 0.35±0.05 | 2177±169 | 327±39 | 76±14 |
| CH-Oe1 | 584±62 | 74.8±7.5 | 0.47±0.14 | 1856±290 | 275±56 | 75±13 |
| DE-Fen | 630±31 | 69.1±3.8 | 0.30±0.02 | 1752±124 | 309±24 | 88±24 |
| DE-RbW | 635±46 | 72.1±10.9 | 0.26±0.03 | 1702±138 | 310±24 | 97±20 |
| DE-Gwg | 507±27 | 60.4±2.1 | 0.40±0.03 | 1422±73.5 | 284±21 | 113±14 |

| | | | | | | |
|---|---|---|---|---|---|---|
| AT-Neu | 496±36 | 53.1±3.7 | 0.33±0.05 | 1574±121 | 353±49 | 99±11 |
| CH-Fru | 503±56 | 59.6±6.2 | 0.36±0.06 | 1701±153 | 361±40 | 106±22 |
| IT-MBo | 407±42 | 61.9±8.6 | 0.62±0.06 | 1326±188 | 222±12 | 174±18 |
| IT-Tor | 394±60 | 68.6±11.2 | 0.72±0.12 | 946±129 | 317±40 | 160±25 |