# Peer review of "Impacts of fertilization on grassland productivity and water quality across the European Alps under current and warming climate: insights from a mechanistic model"

_Biogeosciences, 2020_

## Referee Comment (RC1) · Edoardo Cremonese (Referee) · 22 Sep 2020

This study addresses a relevant topic and could be interesting for a broad range of disciplines. Methods are sound and robust. It's very well written and presented. Below my main comments and a few minor suggestions

Note to editor and authors: I'm the PI of IT-Tor site. I've done my best to try not to be influenced by any "home-bias" in doing the review.

Main comments

l174, l233-236, Tab3 and fig 2. I'd like to see an evaluation of NEE too. GPP is

not directly measured by the eddy covariance technique; it's the result of a model as well. Why did you choose to show LE in fig2? I would have found NEE or GPP more informative. Consider the idea of adding H, GPP, and NEE plots like fig2 at least in S1.

l191-196: carbon and nutrient pools at least in some of the 9 sites (if not in all of them) could be easily obtained by site PIs. I recommend comparing the results of the spin-up exercise with actual values. Which is the fertility range (modeled and observed) of the 9 sites? What would have changed if rather than starting from the results of the spin-up exercise, the fertilization experiment would have been run on actual carbon and nutrient pools values? (cfr l355-360).

fig4 is very important as it gives an idea of how good the model is in simulating structural grassland properties (biomass and LAI) and their temporal dynamics after grass cuts. Unfortunately, the figure is not very clear. It's hard to understand how well the model reproduces interannual variability, absolute values, temporal dynamics around cuts, ... . I acknowledge that it's difficult to find another solution but it could be worth trying. I do not see snow depth data highly informative in this context (e.g. l255-258 can not be depicted from the plot)

A phytosociological or botanical description of the sites is missing. Relevant community and structural differences exist, to my knowledge, at least in some of the sites included in the study. Many of the results you get (e.g. fig4 and fig5) could be seen and commented in the light of species composition and assemblage. To what extent the fact that species composition is not considered in the model could have influenced some of the results? @l404-408 can be articulated in more detail and with a broader perspective.

Minor

l112-114: I suggest using the "official" fluxnet codes IT-Tor, IT-MBo, ... CH-Fru, ... throughout the manuscript

l161: which are the "soil biogeochemistry parameters" considered fixed and homogeneous between sites? which are the site-specific parameters used?

l168 selected for what?

l173 maybe evaluate model performance is better than confirm

l184 what is the reference simulation? was is introduced before?

l185 the implications of this unrealistic assumption must be further discussed and articulated in particular "thus guaranteeing a nutrient application, similarly to fertilization ... "

l189 maybe "flux tower footprint" is better than "below the flux tower"

l243-253 and tab 4. do you get the same picture using measured data?

l270-275 is a reference to fig 5 missing here?

l328-329: see also the previous comment. A more detailed summary of fixed and site-specific parametrization could be useful

l333: "limitations in grass growth and thus LAI at low nitrogen availability" which is the result pointing in this direction?

l342 "temporal drifts". references?

l356 can you be more precise here? How big the differences between modeled management and true local management dates can be?

l843 fig1 and tab 1. IT-Tor coordinates in table 1 are correct but the position shown in fig1 is wrong

l859-867 fig4 I can't find panel references (i.e. a), b), c), ...) in the plot

---

## Referee Comment (RC2) · Stefano Manzoni (Referee) · 1 Oct 2020

Stefano Manzoni (Referee)

stefano.manzoni@natgeo.su.se

The manuscript by Botter et al. presents results from a model of grassland dynamics, focusing on productivity and nutrient losses along gradients of elevation and fertilization intensity. The topic is interesting and suitable for the audience of Biogeosciences. The process-based model adopted for this study (T&C-BG) is also suitable to answer the main research questions - how does grassland productivity changes across sites in the Alps, and how are productivity and nitrate leaching affected by fertilization regimes? The chosen sites span a wide range of climatic, edaphic, and management conditions, and the model setup and fertilization scenarios in combination with the site-to-site variability allow tackling this question. I have a couple of comments aiming at expanding the scope and impact of the work, and several minor suggestions, listed below.

General comments

- The presented analysis is interesting and complete, but I wonder if it would be possible to run a 'climate change' simulation scenario. Higher temperature is expected to increase the ET/precipitation ratio and decrease soil moisture, which might shift nitrogen losses from leaching to denitrification, or might shift the partitioning of mineral nitrogen in favor of plant biomass unless water stress ensues. These interactions (also in relation to fertilization regimes) would nicely complement the current analyses, and they would increase the potential impact of the work. One option could be to simply use climatic conditions from a lower-elevation site to run simulation at a higher-elevation site, or increase temperature (at constant or variable relative humidity) at a given site. These would of course be rather 'theoretical' explorations, but not dissimilar to those set up to test the effects of altered fertilization.

- The metric used to characterize the efficiency of N conversion to biomass is the ratio of harvested N to N concentration in leachate. Typically, agronomic studies define N-use efficiency as a ratio of N in harvested products over N inputs (fertilization, deposition, fixation if N fixers are present). I wonder if such a metric would be more informative. It would allow comparing sites on a N input basis, and values are easily interpreted as 'partitioning coefficients' telling where the N inputs end up in the system.

Specific and technical comments

General: the chemical formula for nitrate is $NO_3^-$, not $NO_3$, so it might be worth adding the superscript minus throughout the manuscript

L15: "unprecedented" sounds a bit an overstatement

L38: how about gaseous losses? Are they important in the nitrogen budgets of these grasslands?

Question 3: this question is rather generic - we know that mechanistic models can provide guidelines, but are these guidelines relevant/applicable? I would actually skip this question altogether, as testing model-based guidelines in the field is outside the scope of the manuscript

L125: check terminology - water leakage or percolation; nutrient leaching (check throughout the manuscript)

L136: it seems that standard meteorological data are enough to drive the model—are eddy flux data necessary?

L146: nutrient leaching

L168: water leakage

L181-182: this sentence is not very clear - what should be accounted for?

L186: is this really unrealistic? Later it is stated that N applications follow grass cutting, so the modelled timing of N addition is right - is it the amount of added N that is "unrealistic"?

Section 2.4: I see the point of running the model at steady state for each fertilization scenario, but I wonder if equilibrium is reached over time scales relevant for management. If the system reaches equilibrium after 500 years (just as an example), then we should perhaps focus on the transient dynamics after fertilization regime is changed - that is, a timescale relevant for management decisions rather than a timescale for ecosystem equilibration

L265-266: are the actual cutting times at the field sites available?

L270-275: I would refer to Figure 5 in this paragraph

L332: check singular/plural "feedbacks... are realistic"

L399 and 446: verb "to take up", not "to uptake"

L432: how is "optimal fertilization level" defined? As shown in Figure 6, there are diminishing returns on N input, but how can an optimum be defined in these monotonically increasing harvested N vs. input N curves?

Figure 1: is the site Torgnon located in Valtournenche (Valle d'Aosta)? If so, please check the position of the site in this map, as it is outside of Valle d'Aosta, further to the south

Figure 3: would it be possible to highlight the growing season periods? What are the soil moisture sensors measuring during the winter, when the soil is frozen? Is it meaningful to compare modelled soil moisture (I assume liquid phase only) with measured values (affected by both liquid and solid phase) when the soil is frozen? I would focus these comparisons on the growing season only

Figures 4-5: I am not sure I understand why biomass data in Figure 4 do not cover the same year(s) as data shown in Figure 5

Figure 6: "kg" not "Kg" in the axis labels; are the markers and lines indicating the median modelled values (shaded areas are explained in the caption, but I missed the explanation of the lines)?

Table 1: would it be possible to add information on site slope/aspect (if not on flat terrain), and soil type?

Table 3: is net radiation modelled (as affected by modelled energy partitioning at the surface?) or used as an input variable?

Table 4: are the mean values based on the periods with available flux data? Would it be worth including plus/minus standard deviation or some measure of the variability?

---

## Author Comment (AC1) · 28 Oct 2020

This study addresses a relevant topic and could be interesting for a broad range of disciplines. Methods are sound and robust. It's very well written and presented. Below my main comments and a few minor suggestions.
We thank the Referee for acknowledging the relevance of the study and generally for appreciating the manuscript.

Note to editor and authors: I'm the PI of IT-Tor site. I've done my best to try not to be influenced by any "home-bias" in doing the review.

**Main comments**

l174, l233-236, Tab3 and fig 2. I'd like to see an evaluation of NEE too. GPP is not directly measured by the eddy covariance technique; it's the result of a model as well. Why did you choose to show LE in fig2? I would have found NEE or GPP more informative. Consider the idea of adding H, GPP, and NEE plots like fig2 at least in S1.
We reported in Figure 2 the pattern of LE because latent heat represents the variable linking hydrology with the energy exchange at the land-surface. We will report the comparison between observed and simulated H, GPP and NEE as supplementary information in the revised version of the manuscript.
Please however note that we spin-up the carbon pools running the soil-biogeochemistry module for 1000 years using average climatic conditions and prescribed litter inputs taken from preliminary simulations with the soil-biogeochemistry module inactive. This solution brings the system to a dynamic equilibrium, while the actual system might not be in equilibrium with regards to carbon. This discrepancy will be reflected in the comparison with NEE, which is the most problematic variable to compare with observations when the history of land-use is not known, which is very often the case, see also our replay below.

l191-196: carbon and nutrient pools at least in some of the 9 sites (if not in all of them) could be easily obtained by site PIs. I recommend comparing the results of the spin- up exercise with actual values. Which is the fertility range (modeled and observed) of the 9 sites? What would have changed if rather than starting from the results of the spin-up exercise, the fertilization experiment would have been run on actual carbon and nutrient pools values? (cfr l355-360).
We thank the Referee for raising this relevant point, which will be included in the discussion. We agree that total carbon and nutrient pools data could be likely included asking the Project's PIs. However, the measured values might not be representative of the entire "footprint" and most important total carbon and nitrogen will not suffice to initialize the 55 pools required by the model. For instance, we would need the subdivision of SOC in mineral associated organic carbon, particulate organic carbon, dissolved organic carbon and also microbial biomass separated in bacteria and fungi components. This might be available in certain experiments (e.g., Cotrufo et al 2020) but not at all sites. However, we will search for total soil C:N ratios at each site and compare them with what is simulated by the model.

fig4 is very important as it gives an idea of how good the model is in simulating structural grassland properties (biomass and LAI) and their temporal dynamics after grass cuts. Unfortunately, the figure is not very clear. It's hard to understand how well the model reproduces interannual variability, absolute values, temporal dynamics around cuts,... . I acknowledge that it's difficult to find another solution but it could be worth trying. I do not see snow depth data highly informative in this context (e.g. l255-258 can not be depicted from the plot)
We think it is important to show the inter-annual variability but we acknowledge that also the intra-annual variability is important. We will show a few dedicated zooms on the supplementary material to highlight

intra-annual patterns. However, we prefer keeping the pattern of snow depth on Figure 4 as we think this is relevant to show the controls on the beginning of the growing season.

A phytosociological or botanical description of the sites is missing. Relevant community and structural differences exist, to my knowledge, at least in some of the sites included in the study. Many of the results you get (e.g. fig4 and fig5) could be seen and commented in the light of species composition and assemblage. To what extent the fact that species composition is not considered in the model could have influenced some of the results? @l404-408 can be articulated in more detail and with a broader perspective. We will add the botanical description where available and expand the comments in the discussion concerning the limitation of not considering multiple species but just an average grassland. However, it is very likely that the botanical heterogeneity effects will be smaller than the current precision in measuring carbon and water fluxes at ecosystem scale.

**Minor**

l112-114: I suggest using the "official" fluxnet codes IT-Tor, IT-MBo, ... CH-Fru, ... throughout the manuscript
We thank the Referee for the suggestion and we will use the Fluxnet codes in all the Fluxnet sites and use the TERENO coding for the German sites DE-Fen, DE-RbW, DE-Gwg.

l161: which are the "soil biogeochemistry parameters" considered fixed and homogeneous between sites? which are the site-specific parameters used?
We will make this distinction clear highlighting in bold the site-specific parameters in Table S1.

l168 selected for what?
We can clarify "selected for all the simulations".

l173 maybe evaluate model performance is better than confirm
We will follow the Referee's suggestion.

l184 what is the reference simulation? was is introduced before?
Reference simulations are meant as the simulations used for evaluating the model performance before running the different fertilization scenarios. We will clarify it in the manuscript.

l185 the implications of this unrealistic assumption must be further discussed and articulated in particular "thus guaranteeing a nutrient application, similarly to fertilization ... "
The assumption of leaving the cut grass on the ground is clearly unrealistic because it opposes the purpose of the grassland management (i.e., producing yields), but it guarantees a nearly closed nutrient cycle, thus performing the same function of fertilizers, or in other words provide the most target fertilization possible. While aiming at testing the model performance on the baseline scenario, we preferred introducing such an hypothesis instead of assuming a fertilization rate for each site to avoid excessive nutrient addition or excessive nutrient starvation or generally to treat all sites equally. Of course, a posteriori, with all the scenarios we run, we could have relaxed the hypothesis and select a given fertilization regime for each site. However, this would have been only possible a posteriori. We will better explain this hypothesis in the manuscript.

l189 maybe "flux tower footprint" is better than "below the flux tower"
We thank the Referee for the suggestion and we will change the text accordingly.

l243-253 and tab 4. do you get the same picture using measured data?
Please refer to the reply to the second main comment.

l270-275 is a reference to fig 5 missing here?
Yes, thank you for the suggestion.

l328-329: see also the previous comment. A more detailed summary of fixed and site-specific parametrization could be useful
We thank the Reviewer for the suggestion and we will introduce a more detailed list.

l333: "limitations in grass growth and thus LAI at low nitrogen availability" which is the result pointing in this direction?
This observation is supported by the results of the scenario analysis (Figure 6a), where for low-N scenarios grass growth was N-limited. We will clearly refer to this result in the revised version of the manuscript.

l342 "temporal drifts". references?
We added the references as suggested and Takruri et al., 2011and Mittelbach et al., 2012.

l356 can you be more precise here? How big the differences between modeled management and true local management dates can be?
Difference will not be too large, but they can be of a week or two in certain years. We refer to management as the combination of manure application and grass cut. We simulate fertilization using a fixed amount of manure as reported in the literature, which is applied on fixed days of the year. Similarly, we assumed fixed days of the year for the grass cuts. In reality, the manure quantity and the days of manure application as well as the days of the grass cut vary from year to year (see SI in Fu et al., 2019). We will clarify this point in the manuscript.

l843 fig1 and tab 1. IT-Tor coordinates in table 1 are correct but the position shown in fig1 is wrong
Thank you for noticing this. We will correct it.

l859-867 fig4 I can't find panel references (i.e. a), b), c), ...) in the plot
Thank you for noticing this. We will correct it.

References

Mittelbach, H., Lehner, I., & Seneviratne, S. I. Comparison of four soil moisture sensor types under field conditions in Switzerland. *Journal of Hydrology, 430–431*, 39–49. https://doi.org/10.1016/j.jhydrol.2012.01.041, 2012.

Fu, J., Gasche, R., Wang, N., Lu, H., Butterbach-Bahl, K., and Kiese, R. : Dissolved organic carbon leaching from montane grasslands under contrasting climate, soil and management conditions, *Biogeochemistry*, 145(1–2), 47–61. https://doi.org/10.1007/s10533-019-00589-y, 2019.

Cotrufo, M. F., Ranalli, M. G., Haddix, M. L., Six, J., & Lugato, E. Soil carbon storage informed by particulate and mineral-associated organic matter. *Nature Geoscience*, 2020.

---

## Author Comment (AC2) · 28 Oct 2020

The manuscript by Botter et al. presents results from a model of grassland dynamics, focusing on productivity and nutrient losses along gradients of elevation and fertilization intensity. The topic is interesting and suitable for the audience of Biogeosciences. The process-based model adopted for this study (T&C-BG) is also suitable to answer the main research questions - how does grassland productivity changes across sites in the Alps, and how are productivity and nitrate leaching affected by fertilization regimes? The chosen sites span a wide range of climatic, edaphic, and management conditions, and the model setup and fertilization scenarios in combination with the site-to-site variability allow tackling this question. I have a couple of comments aiming at expanding the scope and impact of the work, and several minor suggestions, listed below.

We thank the Referee for the appreciation of the manuscript and for the comments which will be used to improve the presentation of the work.

**General comments**

- The presented analysis is interesting and complete, but I wonder if it would be possible to run a 'climate change' simulation scenario. Higher temperature is expected to increase the ET/precipitation ratio and decrease soil moisture, which might shift nitrogen losses from leaching to denitrification, or might shift the partitioning of mineral nitrogen in favor of plant biomass unless water stress ensues. These interactions (also in relation to fertilization regimes) would nicely complement the current analyses, and they would increase the potential impact of the work. One option could be to simply use climatic conditions from a lower-elevation site to run simulation at a higher-elevation site, or increase temperature (at constant or variable relative humidity) at a given site. These would of course be rather 'theoretical' explorations, but not dissimilar to those set up to test the effects of altered fertilization.

We long thought about this addition and we excluded from the first version of the manuscript, as we did not want to make results and discussion excessively long. However, we agree that integrating a climate-change scenario analysis would increase the impact of the study and we therefore consider implementing it in the next version of the manuscript. We would like to integrate at least one climate scenario for each fertilization scenario, built assuming an increase of air temperature of +3°C and an increase of atmospheric $CO_2$ concentration of 300 ppm. As changes in precipitation will be extremely uncertain and likely within historical stochastic variability (e.g., Fatichi et al 2016), we deem that modifying temperature (and consequently vapor pressure) and $CO_2$ might suffice to highlight what can happen in a future climate.

- The metric used to characterize the efficiency of N conversion to biomass is the ratio of harvested N to N concentration in leachate. Typically, agronomic studies define N-use efficiency as a ratio of N in harvested products over N inputs (fertilization, deposition, fixation if N fixers are present). I wonder if such a metric would be more informative. It would allow comparing sites on a N input basis, and values are easily interpreted as 'partitioning coefficients' telling where the N inputs end up in the system.

We thank the Referee for the suggestion, we will compute the N-use efficiency as the ratio between the harvested N and the input N and add this information on top of the currently used metrics.

**Specific and technical comments**

General: the chemical formula for nitrate is NO3-, not NO3, so it might be worth adding the superscript minus throughout the manuscript

Thank you for noticing this. We will change it.

L15: "unprecedented" sounds a bit an overstatement

The unprecedented was referring to the combination of all these different model components but we agree with the suggestion of the Referee and we will delete the term "unprecedented".

L38: how about gaseous losses? Are they important in the nitrogen budgets of these grasslands?
Since we mention gaseous losses in discussion (L360-361) we agree with the Referee that gaseous losses should be mentioned also in the introduction of the study.

Question 3: this question is rather generic - we know that mechanistic models can provide guidelines, but are these guidelines relevant/applicable? I would actually skip this question altogether, as testing model-based guidelines in the field is outside the scope of the manuscript
The analysis provides some useful information which should be taken into considerations by legislators while setting guidelines for management. Under the additional climate-change scenario such results will be hopefully even more relevant. In the discussion, we do not provide actual guidelines but we rather highlight how mechanistic models can account for variability in soil and hydrological purposes not included in the current fixed-threshold guidelines. We will rephrase this third research question in order to clarify that the goal of the study is not to provide actual guidelines, but to explore possible improvement in the methodology that defines guidelines where mechanistic models could have a more important role.

L125: check terminology - water leakage or percolation; nutrient leaching (check throughout the manuscript)
We thank the Reviewer for this observation and we will make sure to be more consistent throughout the manuscript.

L136: it seems that standard meteorological data are enough to drive the model. Are eddy flux data necessary?
Yes, meteorological data are enough to drive the model, eddy flux data are only used to validate it.

L146: nutrient leaching
We will modify accordingly.

L168: water leakage
We will modify accordingly.

L181-182: this sentence is not very clear - what should be accounted for?
We will rephrase to enhance clarity.

L186: is this really unrealistic? Later it is stated that N applications follow grass cutting, so the modelled timing of N addition is right - is it the amount of added N that is "unrealistic"?
The practice of leaving the cut grass on the ground is unrealistic, but the replacement of the amount of exported nutrients contained in the yield, through some "sort of fertilization" is realistic. We will better explain this hypothesis in the revised manuscript.

Section 2.4: I see the point of running the model at steady state for each fertilization scenario, but I wonder if equilibrium is reached over time scales relevant for management. If the system reaches equilibrium after 500 years (just as an example), then we should perhaps focus on the transient dynamics after fertilization regime is changed - that is, a timescale relevant for management decisions rather than a timescale for ecosystem equilibration
We completely agree with the Referee observation. This choice is simply a pragmatic one. Knowing the land-use and management scenarios of last 500-1000 years is impossible almost everywhere and initializing the model with observations is not possible with current data of total bulk C and N only. Even assuming bulk C and N as representative for the model, we will require at least the separation in SOC components and in microbial types. Such limitation is already discussed in the manuscript (L356-358), but we will expand this section. Theoretically, we could run simulations where different scenarios are considered sequentially so to analyze transients rather than equilibrium condition, but such as solution will likely need many more model scenarios and combinations than what we currently present. We think it is out of scope for this manuscript, but it can be an interesting experiment for a future manuscript – the role of transient C and N pools on biogeochemical response.

L265-266: are the actual cutting times at the field sites available?

Unfortunately, not in each site, only for some.

L270-275: I would refer to Figure 5 in this paragraph

We thank the Reviewer for pointing out the lack of the reference to the figure, we will integrate it.

L332: check singular/plural "feedbacks. . . are realistic"

We thank the Reviewer for pointing out this inconsistency. We will correct it.

L399 and 446: verb "to take up", not "to uptake"

We thank the Reviewer for pointing out this mistake. We will correct it.

L432: how is "optimal fertilization level" defined? As shown in Figure 6, there are diminishing returns on N input, but how can an optimum be defined in these monotonically increasing harvested N vs. input N curves?

The optimal fertilization level is better identified by the maximum of the curve in figure 6d. We will better discuss this point it in the manuscript.

Figure 1: is the site Torgnon located in Valtournenche (Valle d'Aosta)? If so, please check the position of the site in this map, as it is outside of Valle d'Aosta, further to the south

We thank the Reviewer for pointing out this inconsistency. We will correct the location on the map.

Figure 3: would it be possible to highlight the growing season periods? What are the soil moisture sensors measuring during the winter, when the soil is frozen? Is it meaningful to compare modelled soil moisture (I assume liquid phase only) with measured values (affected by both liquid and solid phase) when the soil is frozen? I would focus these comparisons on the growing season only

From what we can understand from metadata, observations should be typically referring only to liquid content but in some case readings during "freezing periods" are problematic and lead to spurious values, we rarely see water content dropping even if soil is below zero degrees in the data. As freezing is occurring only in a few sites and for short periods, we compared the total simulated soil moisture (liquid + solid) with the observed "soil moisture". We will highlight the growing season in the plot.

Figures 4-5: I am not sure I understand why biomass data in Figure 4 do not cover the same year(s) as data shown in Figure 5

This incongruence is related to the different data sources for this study. Data for Figure 4 are provided by the managers of flux towers, while Figure 5 is based on published data of lysimeters and they cover a shorter time period. Unfortunately, lysimeter data for the same time period are not available.

Figure 6: "kg" not "Kg" in the axis labels; are the markers and lines indicating the median modelled values (shaded areas are explained in the caption, but I missed the explanation of the lines)?

We thank the Reviewer for pointing out the inconsistency of "Kg". Yes, the lines connect the median values, we will add the explanation in the Figure caption.

Table 1: would it be possible to add information on site slope/aspect (if not on flat terrain), and soil type?

We can add information concerning the soil type, while generally flux towers are placed on a flat area, with no slope to match the theoretical requirements to observe mass and energy fluxes with the eddy covariance system, therefore, the information would be redundant.

Table 3: is net radiation modelled (as affected by modelled energy partitioning at the surface?) or used as an input variable?

Net radiation is definitely modelled as the sum of the contribution of different land surface components. The value observed in the flux-tower is only used for comparison.

Table 4: are the mean values based on the periods with available flux data? Would it be worth including plus/minus standard deviation or some measure of the variability?

Yes, the mean values refer to the periods with available flux data. We thank the Reviewer for the suggestion and we will add the standard deviation of annual values (mean+- standard dev.).

References

Fatichi, S., Ivanov, V.Y., Paschalis, A., Peleg, N., Molnar, P., Rimkus, S., Kim, J., Burlando, P. and Caporali, E. Uncertainty partition challenges the predictability of vital details of climate change. *Earth's Future*, 4: 240-251. doi:10.1002/2015EF000336, 2016.

---

## Author Response (AR1)

This study addresses a relevant topic and could be interesting for a broad range of disciplines. Methods are sound and robust. It's very well written and presented. Below my main comments and a few minor suggestions.

We thank Dr. Edoardo Cremonese for acknowledging the relevance of the study and generally for appreciating the manuscript.

Note to editor and authors: I'm the PI of IT-Tor site. I've done my best to try not to be influenced by any "home-bias" in doing the review.

**Main comments**

l174, l233-236, Tab3 and fig 2. I'd like to see an evaluation of NEE too. GPP is not directly measured by the eddy covariance technique; it's the result of a model as well. Why did you choose to show LE in fig2? I would have found NEE or GPP more informative. Consider the idea of adding H, GPP, and NEE plots like fig2 at least in S1.

We reported in Figure 2 the pattern of LE because latent heat represents the variable linking hydrology with the energy exchange at the land-surface. We added the comparison between observed and simulated H, GPP and NEE as supplementary information in the revised version of the manuscript (Figure S1, Figure S2 and Figure S3).

l191-196: carbon and nutrient pools at least in some of the 9 sites (if not in all of them) could be easily obtained by site PIs. I recommend comparing the results of the spin- up exercise with actual values. Which is the fertility range (modeled and observed) of the 9 sites? What would have changed if rather than starting from the results of the spin-up exercise, the fertilization experiment would have been run on actual carbon and nutrient pools values? (cfr l355-360).

We thank the Referee for raising this relevant point, which was included in the discussion. We agree that total carbon and nutrient pools data could be likely included asking the Project's PIs. However, the measured values might not be representative of the entire "footprint" and most important total carbon and nitrogen will not suffice to initialize the 55 pools required by the model. We searched for total soil C:N ratios at each site and compared them with what is simulated by the model (Table S3). We refer to this analysis in the discussion (L401-412), even though also this comparison is particularly challenging.

fig4 is very important as it gives an idea of how good the model is in simulating structural grassland properties (biomass and LAI) and their temporal dynamics after grass cuts. Unfortunately, the figure is not very clear. It's hard to understand how well the model reproduces interannual variability, absolute values, temporal dynamics around cuts,... . I acknowledge that it's difficult to find another solution but it could be worth trying. I do not see snow depth data highly informative in this context (e.g. l255-258 can not be depicted from the plot)

We think it is important to show the inter-annual variability but we acknowledge that also the intra-annual variability is important. We added zooms of yearly LAI pattern on the supplementary material to highlight intra-annual patterns (Figure S4). We prefer keeping the pattern of snow depth on Figure 4 as we think this is relevant to show how snow cover controls the beginning of the growing season.

A phytosociological or botanical description of the sites is missing. Relevant community and structural differences exist, to my knowledge, at least in some of the sites included in the study. Many of the results you get (e.g. fig4 and fig5) could be seen and commented in the light of species composition and

assemblage. To what extent the fact that species composition is not considered in the model could have influenced some of the results? @l404-408 can be articulated in more detail and with a broader perspective.
We added the botanical description in Table 1 and expanded the comments in the discussion concerning the limitation of not considering multiple species but just an average grassland (L466-468). However, it is very likely that the botanical heterogeneity effects will be smaller than the current precision in measuring carbon and water fluxes at ecosystem scale.

**Minor**

l112-114: I suggest using the "official" fluxnet codes IT-Tor, IT-MBo, ... CH-Fru, ... throughout the manuscript
We thank the Referee for the suggestion and used the Fluxnet codes in all the Fluxnet sites and the TERENO coding for the German sites DE-Fen, DE-RbW, DE-Gwg.

l161: which are the "soil biogeochemistry parameters" considered fixed and homogeneous between sites? which are the site-specific parameters used?
We made this distinction clear highlighting in bold the site-specific parameters in Table S1.

l168 selected for what?
We clarified "selected for all the simulations".

l173 maybe evaluate model performance is better than confirm
We followed the Referee's suggestion and we modified accordingly.

l184 what is the reference simulation? was is introduced before?
Reference simulations are meant as the simulations used for evaluating the model performance before running the different fertilization scenarios. We clarified it in the manuscript (L183-184).

l185 the implications of this unrealistic assumption must be further discussed and articulated in particular "thus guaranteeing a nutrient application, similarly to fertilization ... "
The assumption of leaving the cut grass on the ground is clearly unrealistic because it opposes the purpose of the grassland management (i.e., producing yields), but it guarantees a nearly closed nutrient cycle, thus performing the same function of fertilizers, or in other words provide the most targeted fertilization possible. While aiming at testing the model performance on the baseline scenario, we preferred introducing such a hypothesis instead of assuming a fertilization rate for each site to avoid excessive nutrient addition or excessive nutrient starvation or generally to treat all sites equally. We better explained this hypothesis in the manuscript (L195-201).

l189 maybe "flux tower footprint" is better than "below the flux tower"
We thank the Referee for the suggestion and we changed the text accordingly.

l243-253 and tab 4. do you get the same picture using measured data?
Please refer to the reply to the second main comment.

l270-275 is a reference to fig 5 missing here?
Yes, thank you for the suggestion. We added the reference to Figure 5.

l328-329: see also the previous comment. A more detailed summary of fixed and site-specific parametrization could be useful
We thank the Reviewer for the suggestion and we distinguished the site-specific parametrization highlighting it in bold.

l333: "limitations in grass growth and thus LAI at low nitrogen availability" which is the result pointing in this direction?
This observation is supported by the results of the scenario analysis (Figure 6a), where for low-N scenarios grass growth was N-limited. We clearly referred to this result in the revised version of the manuscript.

l342 "temporal drifts". references?
We added the references as suggested Takruri et al., 2011and Mittelbach et al., 2012.

l356 can you be more precise here? How big the differences between modeled management and true local management dates can be?
Difference will not be too large, but they can be of a week or two in certain years. We refer to management as the combination of manure application and grass cut. We simulate fertilization using a fixed amount of manure as reported in the literature, which is applied on fixed days of the year. Similarly, we assumed fixed days of the year for the grass cuts. In reality, the manure quantity and the days of manure application as well as the days of the grass cut vary from year to year (see SI in Fu et al., 2019). We clarified this point in the manuscript (L397-399).

l843 fig1 and tab 1. IT-Tor coordinates in table 1 are correct but the position shown in fig1 is wrong
Thank you for noticing this. We corrected it.

l859-867 fig4 I can't find panel references (i.e. a), b), c), ...) in the plot
Thank you for noticing this. We added the references to the panels.

The manuscript by Botter et al. presents results from a model of grassland dynamics, focusing on productivity and nutrient losses along gradients of elevation and fertilization intensity. The topic is interesting and suitable for the audience of Biogeosciences. The process-based model adopted for this study (T&C-BG) is also suitable to answer the main research questions - how does grassland productivity changes across sites in the Alps, and how are productivity and nitrate leaching affected by fertilization regimes? The chosen sites span a wide range of climatic, edaphic, and management conditions, and the model setup and fertilization scenarios in combination with the site-to-site variability allow tackling this question. I have a couple of comments aiming at expanding the scope and impact of the work, and several minor suggestions, listed below.
We thank Dr. Stefano Manzoni for the appreciation of the manuscript and for the comments which were used to improve the presentation of the work.

**General comments**

- The presented analysis is interesting and complete, but I wonder if it would be possible to run a 'climate change' simulation scenario. Higher temperature is expected to increase the ET/precipitation ratio and decrease soil moisture, which might shift nitrogen losses from leaching to denitrification, or might shift the partitioning of mineral nitrogen in favor of plant biomass unless water stress ensues. These interactions (also in relation to fertilization regimes) would nicely complement the current analyses, and they would increase the potential impact of the work. One option could be to simply use climatic conditions from a lower-elevation site to run simulation at a higher-elevation site, or increase temperature (at constant or variable relative humidity) at a given site. These would of course be rather 'theoretical' explorations, but not dissimilar to those set up to test the effects of altered fertilization.
We long thought about this addition and we excluded from the first version of the manuscript, as we did not want to make results and discussion excessively long. However, we agree that integrating a climate-change scenario analysis increases the impact of the study and we therefore implemented it in this new version of the manuscript. We integrated one climate scenario for each fertilization scenario, built assuming an increase of atmospheric $CO_2$ concentration of 250 ppm and an increase of air temperature of +3°C. As changes in precipitation will be extremely uncertain and likely within historical stochastic variability (e.g., Fatichi et al

2016), we deem that modifying temperature (and consequently vapor pressure) and $CO_2$ might suffice to highlight what can happen in a future climate, even though we refer to this as "modified climate scenario" as it mostly represents a sensitivity test. The modified climate analysis is integrated in the different sections of the manuscript and we specifically added a new section 3.3 entirely dedicated to results of climate change scenarios Results are summarized in the new Figure 7 and integrated in the already existing Figure 8 (Figure 7 in the previous version of the manuscript).

- The metric used to characterize the efficiency of N conversion to biomass is the ratio of harvested N to N concentration in leachate. Typically, agronomic studies define N-use efficiency as a ratio of N in harvested products over N inputs (fertilization, deposition, fixation if N fixers are present). I wonder if such a metric would be more informative. It would allow comparing sites on a N input basis, and values are easily interpreted as 'partitioning coefficients' telling where the N inputs end up in the system.
We thank the Referee for the suggestion. We computed the N-use efficiency as the ratio between the harvested N and the input N (Figure 1). However, results would not add information compared to the metrics we already used in the manuscript but it would rather be redundant. We therefore decided to exclude this analysis from the manuscript

[Figure]

**Figure 1.** Ratio between harvested N and N input as a function of the percentage of the N input.

**Specific and technical comments**

General: the chemical formula for nitrate is NO3-, not NO3, so it might be worth adding the superscript minus throughout the manuscript
Thank you for noticing this. We changed it throughout the manuscript.

L15: "unprecedented" sounds a bit an overstatement
The unprecedented was referring to the combination of all these different model components but we agree with the suggestion of the Referee and we deleted the term "unprecedented".

L38: how about gaseous losses? Are they important in the nitrogen budgets of these grasslands?
Since we mention gaseous losses in discussion (L360-361) we agree with the Referee that gaseous losses should be mentioned also in the introduction of the study. We added a sentence at L40-42.

Question 3: this question is rather generic - we know that mechanistic models can provide guidelines, but are these guidelines relevant/applicable? I would actually skip this question altogether, as testing model-based guidelines in the field is outside the scope of the manuscript

The analysis provides some useful information, which should be taken into considerations by legislators while setting guidelines for management. In the discussion, we do not provide actual guidelines but we rather highlight how mechanistic models can account for variability in soil and hydrological purposes not included in the current fixed-threshold guidelines. We rephrased this third research question as follows:
(3) Can mechanistic models provide insights for legislators setting guidelines for management also in view of changing climatic conditions?

L125: check terminology - water leakage or percolation; nutrient leaching (check throughout the manuscript)
We thank the Reviewer for this observation and we made sure to be consistent throughout the manuscript.

L136: it seems that standard meteorological data are enough to drive the model. Are eddy flux data necessary?
Yes, meteorological data are enough to drive the model, eddy flux data are only used to validate it.

L146: nutrient leaching
We modified accordingly.

L168: water leakage
We modified accordingly.

L181-182: this sentence is not very clear - what should be accounted for?
We rephrased to enhance clarity (L192-194).

L186: is this really unrealistic? Later it is stated that N applications follow grass cutting, so the modelled timing of N addition is right - is it the amount of added N that is "unrealistic"?
The practice of leaving the cut grass on the ground is unrealistic, but the replacement of the amount of exported nutrients contained in the yield, through some "sort of fertilization" is realistic. We better explained this hypothesis (195-201).

Section 2.4: I see the point of running the model at steady state for each fertilization scenario, but I wonder if equilibrium is reached over time scales relevant for management. If the system reaches equilibrium after 500 years (just as an example), then we should perhaps focus on the transient dynamics after fertilization regime is changed - that is, a timescale relevant for management decisions rather than a timescale for ecosystem equilibration
We completely agree with the Referee observation. This choice is simply a pragmatic one. Knowing the land-use and management scenarios of last 500-1000 years is impossible almost everywhere and initializing the model with observations is not possible with current data of total bulk C and N only. Even assuming bulk C and N as representative for the model, we will require at least the separation in SOC components and in microbial types. Such limitation was already discussed in the manuscript, but we expanded this section also referring to a comparison of soil C:N ratio obtained through spin-up vs the values observed in literature (L401-412, Table S3 and Figure S7). Theoretically, we could run simulations where different scenarios are considered sequentially so to analyze transients rather than equilibrium conditions, but such as solution will likely need many more model scenarios and combinations than what we currently present. We think it is out of scope for this manuscript, but it can be an interesting experiment for a future manuscript – the role of transient C and N pools on biogeochemical response.

L265-266: are the actual cutting times at the field sites available?
Unfortunately, not in each site, only for some.

L270-275: I would refer to Figure 5 in this paragraph
We thank the Reviewer for pointing out the lack of the reference to the figure, we integrated it.

L332: check singular/plural "feedbacks. . . are realistic"
We thank the Reviewer for pointing out this inconsistency. We corrected it.

L399 and 446: verb "to take up", not "to uptake"
We thank the Reviewer for pointing out this mistake. We corrected it.

L432: how is "optimal fertilization level" defined? As shown in Figure 6, there are diminishing returns on N input, but how can an optimum be defined in these monotonically increasing harvested N vs. input N curves?
The optimal fertilization level is better identified by the maximum of the curve in figure 6d. We better discussed this point it in the manuscript.

Figure 1: is the site Torgnon located in Valtournenche (Valle d'Aosta)? If so, please check the position of the site in this map, as it is outside of Valle d'Aosta, further to the south
We thank the Reviewer for pointing out this inconsistency. We corrected the location on the map.

Figure 3: would it be possible to highlight the growing season periods? What are the soil moisture sensors measuring during the winter, when the soil is frozen? Is it meaningful to compare modelled soil moisture (I assume liquid phase only) with measured values (affected by both liquid and solid phase) when the soil is frozen? I would focus these comparisons on the growing season only
From what we can understand from metadata, observations should be typically referring only to liquid content but in some case readings during "freezing periods" are problematic and lead to spurious values, we rarely see water content dropping even if soil is below zero degrees in the data. As freezing is occurring only in a few sites and for short periods, we compared the total simulated soil moisture (liquid + solid) with the observed "soil moisture". We highlight the growing season adding grey background in correspondence of growing season in the plot.

Figures 4-5: I am not sure I understand why biomass data in Figure 4 do not cover the same year(s) as data shown in Figure 5
This incongruence is related to the different data sources for this study. Data for Figure 4 are provided by the managers of flux towers, while Figure 5 is based on published data of lysimeters and they cover a shorter time period. Unfortunately, lysimeter data for the same time period are not available.

Figure 6: "kg" not "Kg" in the axis labels; are the markers and lines indicating the median modelled values (shaded areas are explained in the caption, but I missed the explanation of the lines)?
We thank the Reviewer for pointing out the inconsistency of "Kg". Yes, the lines connect the median values. We corrected the word "Kg" into "kg" and added the explanation of the connecting lines in the Figure caption.

Table 1: would it be possible to add information on site slope/aspect (if not on flat terrain), and soil type?
We added information concerning the soil type, while generally flux towers are placed on a flat area, with no slope to match the theoretical requirements to observe mass and energy fluxes with the eddy covariance system, therefore, the information would be redundant.

Table 3: is net radiation modelled (as affected by modelled energy partitioning at the surface?) or used as an input variable?
Net radiation is definitely modelled as the sum of the contribution of different radiation fluxes. The value observed in the flux-tower is only used for comparison.

Table 4: are the mean values based on the periods with available flux data? Would it be worth including plus/minus standard deviation or some measure of the variability?
Yes, the mean values refer to the periods with available flux data. We thank the Reviewer for the suggestion and we added the standard deviation of annual values (mean+- standard dev.).

References

Mittelbach, H., Lehner, I., & Seneviratne, S. I. Comparison of four soil moisture sensor types under field conditions in Switzerland. *Journal of Hydrology*, *430–431*, 39–49. https://doi.org/10.1016/j.jhydrol.2012.01.041, 2012.

Fatichi, S., Ivanov, V.Y., Paschalis, A., Peleg, N., Molnar, P., Rimkus, S., Kim, J., Burlando, P. and Caporali, E. Uncertainty partition challenges the predictability of vital details of climate change. *Earth's Future*, 4: 240-251. doi:10.1002/2015EF000336, 2016.

Fu, J., Gasche, R., Wang, N., Lu, H., Butterbach-Bahl, K., and Kiese, R. : Dissolved organic carbon leaching from montane grasslands under contrasting climate, soil and management conditions, *Biogeochemistry*, 145(1–2), 47–61. https://doi.org/10.1007/s10533-019-00589-y, 2019.

Takruri, M., Rajasegarar, S., Challa, S., Leckie, C., & Palaniswami, M. Spatio-temporal modelling-based drift-aware wireless sensor networks. *IET Wireless Sensor Systems*, *1*(2), 110–122. https://doi.org/10.1049/iet-wss.2010.0091, 2011.